# Conceptrol: Concept Control of Zero-shot Personalized Image Generation

## Abstract

Personalized image generation with text-to-image diffusion models generates unseen images based on reference image content. Zero-shot adapters such as IP-Adapter and OminiControl are interesting because they do not require test-time fine-tuning. However, they struggle to balance preserving the personalized content and adherence to the text prompt. We identify a design flaw explaining this performance gap: current adapters inadequately integrate reference images with the textual descriptions. The generated images therefore tend to replicate the reference or misunderstand the personalized target. We propose Conceptrol, a simple yet effective framework to enhance zero-shot adapters without computational overhead. Conceptrol constrains the attention of the visual specification with a textual concept mask that improves subject-driven generation capabilities. It achieves as much as 89% improvement on personalization benchmarks over the vanilla IP-Adapter and outperforms fine-tuning approaches like Dreambooth LoRA.

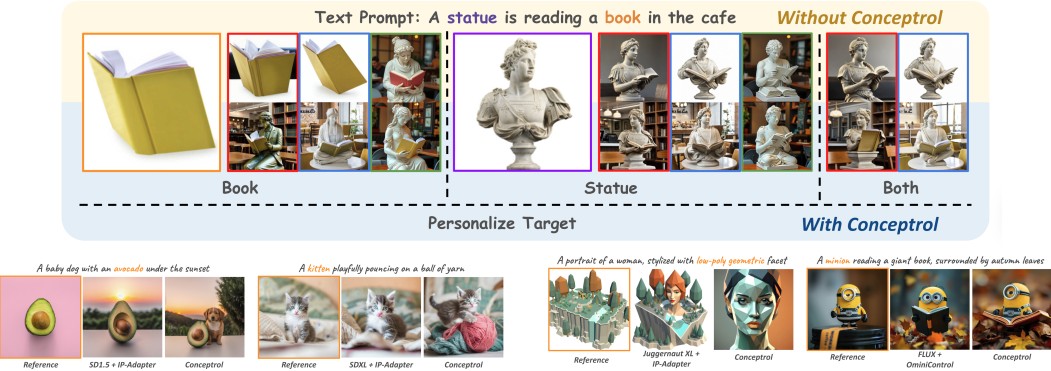

Figure 1: We propose *Conceptrol*, a training-free control method that markedly improves the customization capabilities of zero-shot adapters. Existing adapters (top yellow row) exhibit copy-paste artifacts ( duplicating the book) and mismatched specifications (a red book or an inconsistent statue). Our method *Conceptrol* (blue row) accurately preserves identity while strictly following the text prompt and is applicable to different references (statue and yellow book). It supports different base models (Stable Diffusion, SDXL, FLUX), personalized targets (objects and styles), and parameters (SDXL and Juggernaut XL), without computation overhead, training data, or auxiliary models.

## 1 Introduction

Personalizing image generation with reference images is a significant and practical application of text-to-image diffusion models (Rombach et al., 2022; Podell et al., 2023; Chen et al., 2023; Labs, 2023; Esser et al., 2024). Few-shot fine-tuning approaches like Textual Inversion (Gal et al., 2022) and DreamBooth (Ruiz et al., 2023) fine-tune an off-the-shelf diffusion model with a few reference images to yield a personalized model. However, as model sizes grow, fine-tuning becomes computationally expensive, limiting their efficiency for real-world applications. Another line of work focuses on training additional adapters, such as IP-Adapter (Ye et al., 2023) or OminiControl (Tan

et al., 2024), to enable zero-shot personalization. Adapters are more efficient for inference because there is no need for testing-time fine-tuning. However, they struggle to balance the preservation of the visual specifications while adhering to the textual prompts. Recent research (Tan et al., 2024; Cai et al., 2024) attributes this challenge to a lack of training data that pairs the same reference subject with diverse textual descriptions. Yet even with additional data, zero-shot adapters are weaker in following textual instructions compared to base models without reference images (Tan et al., 2024).

This paper examines the limitations of zero-shot personalization from a design perspective. In personalization, the generated image should (i) remain consistent with the reference's visual specification, and (ii) adhere to the textual instruction. Our key insight is that fine-tuning methods make better use of the textual concept underlying the reference image. For instance, DreamBooth integrates the concept of "cat" into its loss when given reference images of a particular cat, thereby leveraging the base diffusion model's text understanding ability (e.g., Stable Diffusion (Rombach et al., 2022), FLUX (Labs, 2023)). By contrast, zero-shot adapters are unaware of such textual concepts, yielding undesirable outcomes: a direct 'copy-paste' replication (Ye et al., 2023) of the reference (Figure 1), or incorrect personalization, where the provided object is misapplied to the wrong subject (see Figure 2, book to statue).

*Can adapters be guided to explicitly leverage textual concepts without re-training?* To investigate, we present a comprehensive analysis of the attention mechanism in Section 3.2, as it serves as the primary interface for integrating personalized reference images. Our analysis of attention maps leads to three observations: (1) without textual concept constraints, the adapter's attention to reference images often misaligns with the *intended area*, i.e., the region where the text-to-image model seeks to generate the subject; (2) adapters can transfer the appearance of reference images within regions of high attention; and (3) certain attention blocks in text-to-image diffusion models (Rombach et al., 2022; Podell et al., 2023; Labs, 2023) consistently produce a ***textual concept mask***—an attention map that reliably highlights the *intended area* of the textual concept.

Based on insights (1) and (2), we hypothesize that ***visual specifications*** need to be constrained by the *intended area* through attention. Manual attention masking (Marcos-Manchón et al., 2024; Zhou et al., 2024; Wu et al., 2023; Wang et al., 2024b) has been widely explored in text-to-image generation, which typically depends on manually provided regions of interest. However, it has two key limitation for personalization: (i) it requires the user to predict the desired layout of the generated images; (ii) it cannot be used to personalize abstract concept such as style, where the *intended area* does not have clear boundary. Building further open insight (3), we introduce ***textual concept mask*** instead of manual masking, which can be obtained alongside the generation and applicable to capture abstract textual concept such as 'low-poly geometric' or 'pixel art'.

To that end, we introduce a simple but effective method called ***Conceptrol***. Conceptrol enhances the zero-shot customization of existing adapters in a training-free manner. During inference, given a global text prompt (e.g., "a kitten playfully pouncing on a ball of yarn"), a textual concept (e.g., "kitten"), and a visual specification (e.g., an image of a specific kitten) for customized target, we derive a textual concept mask from specific attention blocks in the base models, and use it to constrain the attention of the visual specification. Intuitively, our approach harnesses the original capabilities of text-to-image diffusion models and the appearance-transfer capabilities of lightweight adapters, improving the trade-off between concept preservation and prompt adherence.

We conduct comprehensive quantitative experiments using DreamBench++ (Peng et al., 2024) and human evaluation from MTurk (Amazon Mechanical Turk). We evaluate Conceptrol with IP-Adapter (Ye et al., 2023) for UNet-based diffusion models (e.g. Stable Diffusion, SDXL) and OminiControl (Tan et al., 2024) for DiT-based models (e.g. FLUX). With Conceptrol, these zero-shot adapters make significant improvements by a large margin on personalized image generation benchmarks, even surpassing fine-tuning methods. To summarize, our contributions are as follows:

- We identify a critical design flaw in zero-shot adapters, showing that neglecting textual concepts leads to undesired attention in reference images.

- We introduce the simple yet effective ***Conceptrol***, which automatically extracts a ***textual concept mask*** from attention blocks in the base model to increase the attention of visual specification on the *intended area* of the personalized target while suppressing it on irrelevant regions.

- Extensive evaluations show that Conceptrol improves zero-shot personalized image generation by a large margin and even surpasses fine-tuning methods despite its simplicity.

## 2 PRELIMINARIES

**Text-to-Image Diffusion Models** generate images by gradually denoising a latent variable sampled from a Gaussian distribution. The denoising process is guided by a neural network (often a UNet (Ronneberger et al., 2015; Rombach et al., 2022; Podell et al., 2023) or a ViT (Dosovitskiy et al., 2020; Labs, 2023)) and conditioned on a text prompt. Starting from random noise, the model iteratively refines the latent until reaching a clean representation, which is then decoded into an image using a VAE (Kingma, 2013). Recent models such as FLUX (Labs, 2023) adopt flow matching (Lipman et al., 2022), which can be viewed as a variant of the diffusion framework (Wang et al., 2024a). We provide the detailed formulation in the Appendix D.

**Terminologies used in Conceptrol**. Let $c_{\text{image}}$ denote the *visual specification*, which corresponds to the provided reference image. Let $c_{\text{text}}$ denote the *text condition*, a representation corresponding to the text description. Formally, $c_{\text{text}}$ is represented as $c_{\text{text}} \in \mathbb{R}^{N \times C}$, where $N$ is the number of tokens in the text and $C$ is the dimensionality of each token. We further introduce the notion of a *textual concept*, $c_{\text{concept}}$, defined as a substring of $c_{\text{text}}$. For instance, in the prompt *"A dog is playing with a cute cat in the living room,"* the textual concept can be *"a cute cat"* if the visual specification provided represents a cat. The textual concept $c_{\text{concept}}$ is sliced from $c_{\text{text}}$ as $c_{\text{concept}} = c_{\text{text}}[i_s : i_e, :] \in \mathbb{R}^{N' \times C}$, where $[i_s, i_e]$ are the start and end indices (excluded) of the textual concept within $c_{\text{text}}$. Here, $N' = i_e - i_s$ denotes the number of tokens representing the textual concept.

**IP-Adapter for U-Net Models: Direct Adding.** UNet models (Saharia et al., 2022; Podell et al., 2023) incorporate conditions using cross-attention. For a noisy latent $x_t$, a condition $c$, and matrices $W_q^{(l)}$, $W_k^{(l)}$, and $W_v^{(l)}$ of attention block $l$, the cross-attention is computed as:

$$\text{Attn}^{(l)}(x_t, c) = \underbrace{\text{Softmax}\left(\frac{(W_q^{(l)}x_t)(W_k^{(l)}c)^T}{\sqrt{d}}\right)}_{A_c^{(l,x_t)}} W_v^{(l)}c, \tag{1}$$

where $d$ is the dimensionality of the query and key vectors, $A_c^{(l,x_t)}$ is the attention map. In text-to-image generation, UNet models apply Equation 1 with a text condition $c_{\text{text}}$ as $c$. The IP-Adapter extends this mechanism by introducing an additional cross-attention for the reference image represented by CLIP embedding, $c_{\text{image}}$. The combined attention $\text{Attn}_{\text{IP}}^{(l)}$ is defined as:

$$\text{Attn}_{\text{IP}}^{(l)}(x_t, c_{\text{text}}, c_{\text{image}}; \lambda) = \text{Attn}^{(l)}(x_t, c_{\text{text}}) + \lambda \cdot \text{Attn}^{(l)}(x_t, c_{\text{image}}). \tag{2}$$

Above, $\lambda$ represents the *IP Scale*, a weighting hyperparameter on the reference image.

**OminiControl for DiT Models: MM-Attention.** DiT models use multi-modal attention to fuse text conditions to the latent. OminiControl further concatenates image tokens. Given a noisy latent $x_t$, text tokens $c_{\text{text}}$ and image tokens $c_{\text{image}}$, the fused latent is represented as $x_t' = [c_{\text{text}}, x_t, c_{\text{image}}]$, where $[\cdot, \cdot, \cdot]$ denotes token concatenation. Then the attention is computed as:

$$\text{Attn}_{\text{omini}}^{(l)}(x_t, c_{\text{text}}, c_{\text{image}}; \lambda) = \underbrace{\text{Softmax}\left(\frac{(W_q^{(l)}x_t')(W_k^{(l)}x_t')^T}{\sqrt{d}} + B(\lambda)\right)}_{A_{mm}^{(l,x_t)}} W_v^{(l)}x_t'. \tag{3}$$

where $B(\lambda)$ is used to control the conditioning scale of image and $A_{mm}^{(l,x_t)}$ is the fused attention map. Given $c_{\text{text}} \in R^{M \times d}$, $x_t, c_{\text{image}} \in R^{N \times d}$, it is computed as:

$$B(\lambda) = \begin{bmatrix} \mathbf{0}_{M \times M} & \mathbf{0}_{M \times N} & \mathbf{0}_{M \times N} \\ \mathbf{0}_{N \times M} & \mathbf{0}_{N \times N} & \log(\lambda)\mathbf{1}_{N \times N} \\ \mathbf{0}_{N \times M} & \log(\lambda)\mathbf{1}_{N \times N} & \mathbf{0}_{N \times N} \end{bmatrix}. \tag{4}$$

In Eqs. 3 and 4, a larger $\lambda$ increases the scaling of the image conditioning. Compared to IP-Adapter, OminiControl uses a larger dataset of pairs featuring the same subjects with diverse text prompts.

## 3 METHODOLOGY

In this section, we mainly analyze two typical zero-shot adapters including IP-Adapter (Ye et al., 2023), which uses direct adding for UNet-based model, and OminiControl (Tan et al., 2024), which uses MM-Attention for DiT-based model through the lens of attention mechanism.

### 3.1 WHY DO WE NEED TO CONSTRAINT VISUAL SPECIFICATION WITH TEXTUAL CONCEPT?

Current zero-shot adapters (Ye et al., 2023; Tan et al., 2024) treat reference images symmetrically with the entire textual instruction, i.e., exchanging $c_{\text{text}}$ and $c_{\text{image}}$ does not alter the modeling. This design leads to two undesirable consequences. First, without explicit decoupling, conflicts can arise between the two conditions. For example, with a book as the reference image and the textual prompt "A statue is reading the book," the adapter may mistakenly personalize the statue rather than the book. As shown in Figure 2 (row 1), a low image conditioning strength (IP Scale) in IP-Adapter (Ye et al., 2023) fails to preserve the concept, while increasing the scale causes deviations from the prompt and produces copy-paste artifacts.

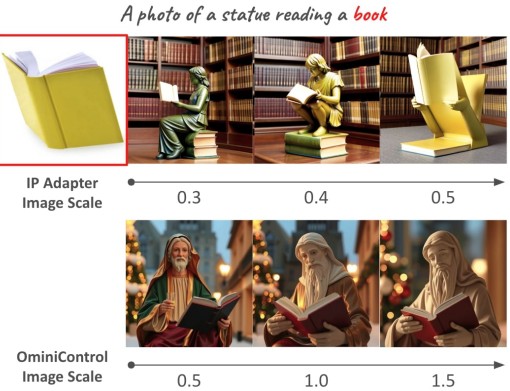

Figure 2: **Treating image conditions equivalent to textual instruction can lead to ambiguity.** The first row shows IP-Adapter on Stable Diffusion 1.5 with varying IP Scales, where increasing scale shifts the output from "a statue reading a book" to "a book statue." The second row shows OminiControl on FLUX failing to preserve the color of the book as yellow, instead generating red books at different conditioning scales.

Secondly, symmetric modeling requires the model to resolve ambiguities and conflicts directly learned from the dataset. This makes it challenging to construct a dataset that is both sufficiently diverse and capable of covering the full range of personalized targets. Even in OminiControl (Tan et al., 2024), which is trained on data pairs of the same subject with different text prompts, the model still struggles to identify which subject in the prompt should be customized. For example, when generating "A statue is reading a book" with reference images of a specific book, the model may incorrectly prioritize "statue" over "book," effectively ignoring the reference image, as shown in Figure 2 (row 2).

Therefore, instead of treating the reference image and text prompt equally, it's required to constraint the visual specification with a particular textual concept. For instance, in the prompt "A photo of a statue reading a book", the reference image should only influence the generation of "a book". Otherwise, the reference might interfere the generation of "statue" as well and lead to artifacts.

### 3.2 WHAT DOES THE ATTENTION OF NOISY LATENT TO THE CONDITION INDICATE?

Since the attention block is the primary interface through which both IP-Adapter (Ye et al., 2023) and OminiControl (Tan et al., 2024) incorporate additional image conditions, we investigate how these conditions interact with the noisy latent and influence the generation process. Prior works have analyzed attention maps after the full generation (Marcos-Manchón et al., 2024; Zhou et al., 2024) without reference images. In this section, on the contrary, we are interested in the behavior of the attention map *during the diffusion process*, and how the reference images influence it.

To explore these questions, we first analyze the diffusion process without reference images by setting the conditioning scale to zero, but still computing the attention map for the reference image. Next, we use LangSAM (Medeiros, 2024), a segmentation tool based on SAM (Kirillov et al., 2023), to obtain pseudo masks of the customization target. For instance, in Figure 4, (b) shows the mask produced by LangSAM, while (c) displays one of the attention maps. This mask explicitly marks the *intended area* of the base model to generate the corresponding subject. By calculating the AUC between the attention map and the target mask, we quantitatively assess whether the attention map correctly highlights the *intended area*. We provide more details in the Appendix A.

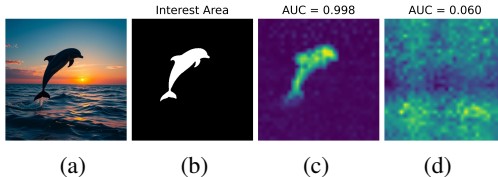

(a)     (b)     (c)     (d)

Figure 4: **Some attention maps of textual concept strongly indicate the interest area.** Shown are examples from the FLUX model, including (a) generated results for "dolphin", (b) segmentation from SAM indicating the subject, and an attention map with *"dolphin"* from (c) `BLOCK 18` and (d) `BLOCK 11`.

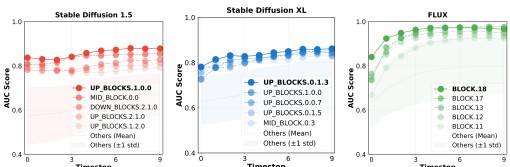

Figure 5: **The attention map of a textual concept directly reveals the regions of interest across various architectures.** We present the AUC scores of the attention maps for the textual concept from different blocks, compared to the ground truth mask, across timesteps in Stable Diffusion 1.5, SDXL, and FLUX.

**Attention to the reference image is misaligned with the intended area.** Figure 3 shows an example of this discrepancy. In this example, the attention map to the text "avocado" closely matches the ground-truth mask of the avocado in the generated result, whereas the attention map to the reference image highlights the dog. Quantitatively, the highest AUC of the image condition's attention maps across all blocks is only 0.38, compared to an AUC of up to 0.99 for the text (e.g., "avocado"). This supports our insights in Section 1 that the attention maps to reference images are not aligned with the intended area of the text-to-image model to generate the corresponding subject.

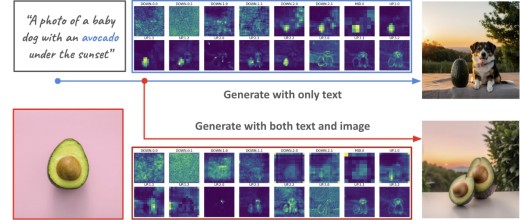

Figure 3: **Incorrect attention map of image conditions.** This example illustrates IP-Adapter results with fully text-based input and with additional image condition added at the 10th of 50 total denoising steps. The blue box shows the attention map of 'avocado,' while the red box highlights the image condition, which incorrectly focuses on the dog area as well, distorting results and reducing text prompt adherence.

**Visual specifications can be transferred within regions of high attention score**. It has been shown that IP-Adapter can transfer visual specification by manually applying an attention mask (Ye et al., 2023). We further verify this in FLUX with OminiControl. Specifically, we first generate the image with only the textual instruction, and obtain the mask of personalized subject. When generating an image with both the textual instruction and reference images, we use it to mask the attention to the reference image to get another generated result. Then we segment from the new result again, and compare with the original masking. The AUC can be as high as 0.9 with more details in the Appendix A. This indicates that adapters can transfer the appearance of reference images within regions of high attention scores.

**Concept-specific attention blocks for text conditions indicate the intended area during generation.** Unlike previous post-hoc analyses of attention maps (Marcos-Manchón et al., 2024), we investigate their characteristics *during* the generation process. For architectures that incorporate cross-attention or multi-modal attention to introduce text conditions, we observe specific blocks that clearly highlight the intended area, as shown in Figure 4. To quantify this, we compare the AUC between attention maps at each block and timestep with the mask obtained via LangSAM. As illustrated in Figure 5, attention maps from these blocks strongly indicate the intended area. These concept-specific blocks include the following: `UP BLOCK 1.0.0` in Stable Diffusion, `UP BLOCK 0.1.3` in SDXL, and `BLOCK 18` in FLUX. We refer to the maps from these specific blocks as ***textual concept masks***, as they directly indicate the intended area of the textual concept.

## 4   CONCEPTROL: CONTROL PERSONALIZATION WITH TEXTUAL CONCEPT

We propose a simple yet effective method to consistently boost the personalization ability of the zero-shot adapter, which we refer to as ***Conceptrol***. Building on the insights, Conceptrol employs a ***textual concept mask*** to adjust the attention map of image conditions so that the intended area receives the highest score, enabling the adapters to accurately transfer the ***visual specification***.

**Conceptrol on Direct Adding**. For the concept-specific attention block $l^*$, such as `UP BLOCK 0.1.3` in SDXL, its attention map at inference timestep $t$ is obtained as $A_{c_{\text{text}}}^{(l^*, x_t)} \in R^{h \times (H \times W) \times M}$, where $h$ is the number of heads, $H, W$ is the size of the feature map and $M$ is the number of text tokens. We slice out the attention map of textual concept $A_{c_{\text{concept}}}^{(l^*, x_t)} = A_{c_{\text{text}}}^{(l^*, x_t)}[:, :, i_s : i_e]$. $A_{c_{\text{concept}}}^{(l^*, x_t)}$ is averaged over the heads and textual concept tokens and normalized to obtain $\widetilde{A}_{c_{\text{concept}}}^{(l^*, x_t)} \in R^{(H \times W)}$:

$$\widetilde{A}_{c_{\text{concept}}}^{(l^*, x_t)} = \text{mean}(A_{c_{\text{concept}}}^{(l^*, x_t)}, \dim = \{0, 2\}), \quad \widetilde{A}_{c_{\text{concept}}}^{(l^*, x_t)} := \frac{\widetilde{A}_{c_{\text{concept}}}^{(l^*, x_t)}}{\max(\widetilde{A}_{c_{\text{concept}}}^{(l^*, x_t)})} \tag{5}$$

During inference, the IP-Adapter's cross-attention can be modified by masking the attention with the image condition:

$$\text{Attn}_{\text{Conceptrol+IP}}^{(l)}(x_t, c_{\text{text}}, c_{\text{image}}, \widetilde{A}_{c_{\text{concept}}}^{(l^*, x_t)}) = \text{Attn}^{(l)}(x_t, c_{\text{text}}) + \lambda \cdot \widetilde{A}_{c_{\text{concept}}}^{(l^*, x_t)} \odot \text{Attn}^{(l)}(x_t, c_{\text{image}}) \tag{6}$$

Where $\odot$ corresponds to element-wise multiplication on the spatial feature.

**Conceptrol on MM-Attention**. Similar to Conceptrol on Direct Adding, given concept-specific attention block $l^*$ (`BLOCK 18` in FLUX), we compute its attention map firstly as $A_{mm}^{(l^*, x_t)} \in R^{(h \times (M+2N) \times (M+2N))}$, where $h$ is the number of heads, $M$ is the number of text tokens, $N$ is the number of latent tokens, at inference timestep $t$ with concatenated tokens $[c_{\text{text}}, x_t, c_{\text{image}}]$. We then slice the attention map of noisy latent with textual concept by: $A_{c_{\text{concept}}}^{(l^*, x_t)} = A_{mm}^{(l^*, x_t)}[:, M : M + N, i_s : i_e]$. Similar to Equation 5, we average over each head and text tokens in textual concept $\widetilde{A}_{c_{\text{concept}}}^{(l^*, x_t)}$ but normalizing it by its mean value.

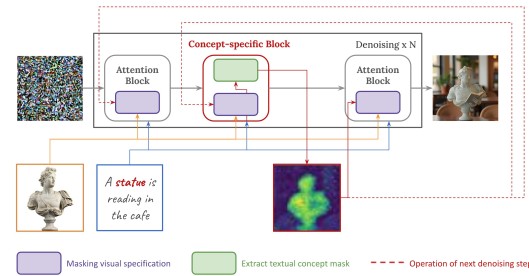

Figure 6: **Overview of Conceptrol**. Conceptrol extracts a textual concept mask highlighting the intended area for a textual concept (e.g., "statue"), from a concept-specific block (e.g., `UP BLOCK 0.1.3` in SDXL). It then adjusts the attention to the corresponding visual specification (i.e., a personalized image of the statue) in the adapters to enhance personalization.

Different from Direct Adding, MM-Attention enforces attention between text and image conditions as well. To further constrain the influence of the image condition on irrelevant concepts, we define mask $M'(\lambda) \in R^{(M \times N)}$ as:

$$M'_{ij}(\lambda) = \log(\epsilon) + \mathbf{1}_{\{i_s \le i < i_e\}} \left[\log(\lambda) - \log(\epsilon)\right] \tag{7}$$

Here $\lambda$ is the conditioning scale of reference images and $\epsilon$ is a value close to 0 such that $\log(\epsilon)$ is extremely small to prevent irrelevant text tokens paying attention to the reference image. Then we alter the attention in OminiControl as:

$$\text{Attn}_{\text{Conceptrol+Omini}}(x_t, c_{\text{text}}, c_{\text{image}}, \widetilde{A}_{c_{\text{concept}}}^{(l^*, t)}) = \text{Softmax}\left(\frac{(W_q x'_t)(W_k x'_t)^T}{\sqrt{d}} + B_{c_{\text{concept}}}(\lambda)\right) W_v x'_t. \tag{8}$$

where $B_{c_{\text{concept}}}(\lambda)$ is computed as:

$$B_{c_{\text{concept}}}(\lambda) = \begin{bmatrix} \mathbf{0}_{M \times M} & \mathbf{0}_{M \times N} & M'(\lambda) \\ \mathbf{0}_{N \times M} & \mathbf{0}_{N \times N} & \log\left(\lambda \widetilde{A}_{c_{\text{concept}}}^{(l^*, x_t)}\right) \cdot \mathbf{1}_{1 \times N} \\ \mathbf{0}_{N \times M} & \log(\lambda) \mathbf{1}_{N \times N} & \mathbf{0}_{N \times N} \end{bmatrix} \tag{9}$$

Here '$\cdot$' represents matrix multiplication. We provide more details in the Appendix C.

**Conceptrol Warmup**. We notice that the attention map is less informative in the early stage to indicate the intended, as shown in Figure 3 and Figure 5. Therefore, instead of starting control initially, we introduce another hyperparameter, **conditioning warmup ratio** $\epsilon$, to prohibit the injection of

"A portrait of a cat in *impressionist* style"   "A portrait of a cat in *Peredvizhniki* style"

Figure 7: **Qualitative examples of personalizing abstract concept.** Conceptrol enables style customization by leveraging concept-specific attention blocks that capture complex, spatially varying masks, which are difficult to specify manually. Unlike IP-Adapter, which often transfers layout information from the reference, Conceptrol focuses on transferring the underlying stylistic attributes without imposing the source layout.

Table 1: **Main Results of Dreambench++.** We follow the default evaluation protocol of Dreambench++ to assess the concept preservation and prompt the following scores of various methods, including both fine-tuning approaches and zero-shot adapters (denoted with *). The best results are highlighted with **underline**. Conceptrol significantly enhances the performance of zero-shot adapters, even surpassing fine-tuning methods like Dreambooth across multiple base models.

| Method | Concept Preservation (CP) | | | | | Prompt Following (PF) | | | | CP · PF |
|---|---|---|---|---|---|---|---|---|---|---|
| | Animal | Human | Object | Style | Overall | Photorealistic | Style Transfer | Imaginative | Overall | **Final Score** |
| Textual Inversion SD | 0.501 | 0.372 | 0.308 | 0.358 | 0.381 | 0.679 | 0.698 | 0.440 | 0.632 | 0.241 |
| DreamBooth SD | 0.646 | 0.196 | 0.491 | 0.476 | 0.496 | 0.789 | 0.778 | 0.510 | 0.723 | 0.359 |
| BLIP SD* | 0.676 | 0.556 | 0.469 | 0.508 | 0.548 | 0.578 | 0.518 | 0.300 | 0.496 | 0.272 |
| IP-Adapter SD* | 0.917 | 0.825 | 0.858 | 0.931 | 0.881 | 0.277 | 0.235 | 0.163 | 0.238 | 0.210 |
| + Conceptrol (Ours)* | 0.644 | 0.397 | 0.433 | 0.389 | 0.500 | 0.904 | 0.759 | 0.594 | 0.795 | **0.397** (+89.0%) |
| DreamBooth LoRA SDXL | 0.751 | 0.310 | 0.544 | 0.713 | 0.597 | 0.896 | 0.898 | 0.752 | 0.865 | 0.517 |
| Emu2 SDXL* | 0.665 | 0.551 | 0.447 | 0.443 | 0.526 | 0.729 | 0.730 | 0.558 | 0.691 | 0.364 |
| IP-Adapter SDXL* | 0.902 | 0.840 | 0.762 | 0.914 | 0.835 | 0.502 | 0.386 | 0.282 | 0.414 | 0.346 |
| +Conceptrol (Ours)* | 0.746 | 0.672 | 0.571 | 0.726 | 0.658 | 0.860 | 0.828 | 0.616 | 0.796 | **0.524** (+51.4%) |
| OmniControl FLUX* | 0.555 | 0.182 | 0.476 | 0.306 | 0.438 | 0.945 | 0.920 | 0.821 | 0.910 | 0.398 |
| +Conceptrol (Ours)* | 0.642 | 0.344 | 0.612 | 0.344 | 0.556 | 0.907 | 0.866 | 0.781 | 0.866 | **0.481** (+20.9%) |

image condition before a preset time step $T' = \epsilon T$, where $T$ is the total inference timesteps. At each timestep $t > T'$, for those block ahead of $l^*$, we use $\widetilde{A}_{C_{\text{concept}}}^{(l^*, x_{t-1})}$ accordingly. Otherwise, we use $\widetilde{A}_{C_{\text{concept}}}^{(l^*, x_t)}$ to control the visual specification. Notice that $\epsilon$ can be adaptively determined by the convergence of the attention mask as well, which offers slightly better results with more computation overhead.. More details can be found in the Appendix C.

**Discussion on the difference between standard attention masking and semantic control**. Standard attention masking and semantic-control methods rely on *external* or *post-hoc* spatial supervision, such as user-defined masks, segmentation maps, or masks obtained after inversion, which assumes the target region is known in advance. These approaches suit editing but do not generalize to zero-shot personalization, where the **concept's location is unknown or abstract**. In contrast, Conceptrol derives an *intended* concept mask directly from the model's **internal concept-specific attention blocks on the fly**, revealing where the model plans to render the textual concept, even for an abstract concept, as Fig. 7 shows. This enables training-free, semantically aligned conditioning that cannot be achieved through standard masking or prior semantic-control techniques.

# 5 EXPERIMENTS

## 5.1 EXPERIMENT SETUP

**Compared methods**. To assess the performance, we systematically compare Conceptrol with other state-of-the-art methods such as Textual Inversion (Gal et al., 2022), DreamBooth (Peng et al., 2024), BLIP Diffusion (Li et al., 2024), and Emu2 (Sun et al., 2024). To access the applicability of Conceptrol across different base models, we apply it with IP-Adapter on UNet-based models, including Stable Diffusion 1.5 and SDXL, and OmniControl on DiT-based model FLUX.

**Evaluation Protocol**. We adhere to the evaluation protocol outlined in DreamBench++ (Peng et al., 2024), a comprehensive dataset for personalized image generation. This benchmark systematically assesses customization performance in terms of concept preservation and prompt following using GPT4 (OpenAI et al., 2024), demonstrating superior alignment with human preferences compared

to other benchmarks (Peng et al., 2024; Gal et al., 2022). For formal evaluation, personalized generation is formulated as a Nash Bargaining problem (Nash et al., 1950). The target is to maximize the Nash utility, the multiplication of concept preservation, and prompt adherence. We further verify Conceptrol on CustomConcept101 (Kumari et al., 2023) with more details in the Appendix.C.

**Human Study**. We also conducted a human study using Amazon Mechanical Turk (MTurk) (Amazon Mechanical Turk) to verify that our method aligns with human preferences. Specifically, participants were presented with pairs of images and asked to select the one that better preserved the original concept and adhered to the prompt. More details are in Appendix B.

**Implementation Details**. We use the recommended settings for the compared methods, including guidance scale, number of denoising steps, and conditioning scale, from their original paper or Dreambench++. For our Conceptrol with IP-Adapter on Stable Diffusion 1.5 and SDXL, we use conditioning scale $\lambda$ as 1.0 and conditioning warmup ratio $\epsilon$ as 0.2; with OminiControl on FLUX, we use the conditioning scale as 1.0 and conditioning warmup ratio as 0.

## 5.2 MAIN RESULTS

We present the main results across different methods with various personalized targets in Table 1.

**Freely elicit potential of existing adapters**. With simple control, we can boost the performance of zero-shot adapters on Stable Diffusion 1.5, SDXL, and FLUX by a large margin. Notably, with Conceptrol, the performance of zero-shot adapters can even surpass fine-tuning methods such as Dreambooth LoRA ($0.397 > 0.359$ on SD 1.5, $0.524 > 0.517$ on SDXL), indicating the potential of these zero-shot adapters can be elicited with negligible computation overhead as shown in Figure 8. Meanwhile, all these improvement is achieved with negligible computation overhead. We provide more details in the Appendix B.

**Improvement over human preference**. We report human study results in Figure 14. In contrast to the results obtained using GPT-4 evaluation, our method performed similarly to the vanilla IP-Adapter on SD and SDXL in terms of concept preservation, while demonstrating significantly better prompt adherence. This observation is consistent with Dreambench-Plus (Peng et al., 2024), which reports that human alignment in GPT-4 evaluation is higher for prompt adherence than for concept preservation. We provide more discussion regarding this observation in Appendix B. On FLUX with OminiControl, Conceptrol can improve concept preservation without losing prompt following. Overall, this result demonstrates the effectiveness of Conceptrol, which enhances prompt adherence or concept preservation without sacrificing the other. We further include a human-preference comparison between Conceptrol and DreamBooth LoRA (both based on SDXL) in Appendix B. The results show that Conceptrol achieves stronger concept preservation while maintaining competitive prompt-following ability.

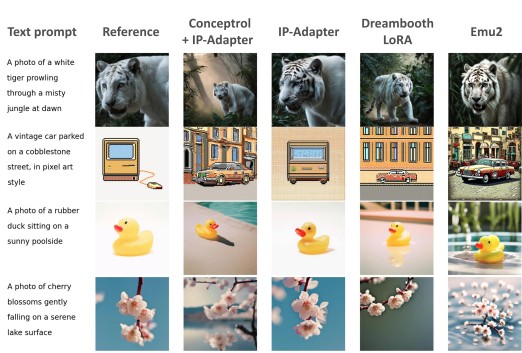

Figure 8: **Qualitative Results on SDXL.** Compared to the vanilla IP-Adapter, Conceptrol enhances customization across diverse targets and becomes competitive with Dreambooth LoRA, which requires fine-tuning. More qualitative results are in Appendix B.

## 5.3 ABLATION STUDY

We systematically evaluate the impact of each component in our methods, including the masking mechanism, conditioning scale, and warm-up ratio, on the personalization score.

**Conditioning Scale**. The conditioning scale defines the default trade-off between concept preservation and prompt adherence in zero-shot adapters. We compare the original IP-Adapter and its variant under Conceptrol, as shown in Figure 10 (a). For both approaches, increasing the conditioning scale enhances concept preservation and diminishes prompt adherence. Notably, Conceptrol achieves a better multiplied score across different conditioning scales.

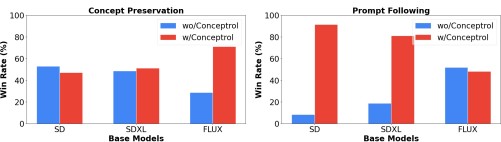

Figure 9: **Human Study Results**. On SD and SDXL with IP-Adapters, Conceptrol boosts performance on prompt following while achieving similar performance of concept preservation. On FLUX with OminiControl, Conceptrol increases performance on concept preservation with similar prompt following.

**Conditioning Warm-Up Ratio**. This ratio is another important hyperparameter; results are shown in Figure 10 (b). The prompt-following score improves as the warmup ratio increases, whereas the concept preservation score decreases. However, Conceptrol consistently improves multiplied scores under each setting. We set the warmup ratio to 0.2 for the IP-Adapter to enhance the prompt following. In terms of OmniControl, we set the warmup ratio to 0.0 since the textual concept mask of FLUX converges faster. We present more details in Appendix C.

**Masking Mechanism**. To evaluate the effectiveness of the textual concept mask, we compare it with three alternative settings: 1) *Non-specific mask*, where the attention mask is directly transferred from the textual concept in each block individually, without using the concept-specific attention block; 2) *Mask from other blocks*, such as DOWN.0.0.0; and 3) *Oracle mask*, where an image is first generated entirely based on the text prompt, followed by segmentation of the subject using SAM (Kirillov et al., 2023) to extract the mask. As shown in Table 2, the textual concept mask outperforms the non-specific mask and masks extracted from uninformative attention blocks such as DOWN.0.0.0. Notably, without additional computational overhead or reliance on auxiliary models, the textual concept mask is as competitive as the oracle mask. Additionally, it can be used to personalize abstract concepts, such as style, while the oracle mask of these abstract concepts remains intractable.

Table 2: **Ablation study on the masking mechanism with SDXL.** CP = concept preservation, PF = prompt following, NFE = number of denoising steps. Our textual concept mask elicits the potential of a text-to-image model and achieves competitive results compared to methods requiring double computation and large segmentation models.

| Masking method | NFE | CP | PF | CP · PF |
|---|---|---|---|---|
| non-specific mask | 50 | 0.670 | 0.611 | 0.409 |
| mask from DOWN.0.0.0 | 50 | 0.611 | 0.582 | 0.356 |
| **textual concept mask (Ours)** | 50 | 0.658 | 0.796 | **0.524** |
| oracle mask (with SAM) | 100 | 0.697 | 0.754 | 0.526 |

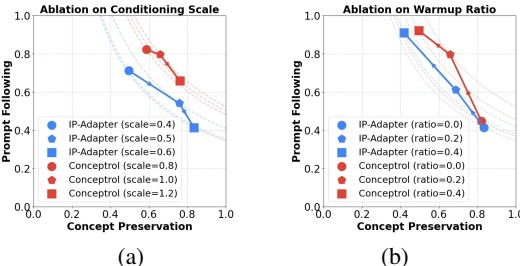

(a)                              (b)

Figure 10: **Ablation study on (a) conditioning scale and (b) warmup ratio using SDXL.** Each dashed line represents the curve $y = \frac{t}{x}$ across the corresponding data points, where a higher curve in the plot indicates better results for personalization. Conceptrol achieves a superior trade-off curve across both hyperparameters.

# 6 RELATED WORK

**Text-to-Image Diffusion Models** have achieved state-of-the-art image generation quality (Rombach et al., 2022; Podell et al., 2023; Labs, 2023; Esser et al., 2024) using text prompts and training on massive, diverse datasets like LAION-5B (Schuhmann et al., 2022). Previous models like Imagen (Saharia et al., 2022), Stable Diffusion 1.5 (Rombach et al., 2022), and SDXL (Podell et al., 2023) used a UNet architecture (Ronneberger et al., 2015). More recently, the Diffusion Transformer (DiT) (Peebles & Xie, 2023) with flow matching (Lipman et al., 2022) has become the dominant design in models such as Stable Diffusion 3 (Esser et al., 2024) and FLUX (Labs, 2023). Both kinds of models integrate text conditions with the noisy latent through attention mechanisms.

**Personalized Image Generation** creates images from text prompts and reference images that depict previously unseen, customized targets (Gal et al., 2022; Ruiz et al., 2023; Ye et al., 2023; Tan et al., 2024). A key challenge in this area is to preserve the novel concept while ensuring consistency with the text prompt, as highlighted by the subject-driven generation benchmark DreamBenchPlus (Peng et al., 2024). It can be formulated as a multi-task problem, requiring concept preservation and prompt adherence, and is analogous to maximizing a Nash welfare (Nash et al., 1950).

**Fine-tuning Methods** adapt diffusion models given personalized content for further generation. Textual Inversion (Gal et al., 2022) freezes all model parameters and tunes only the text embedding

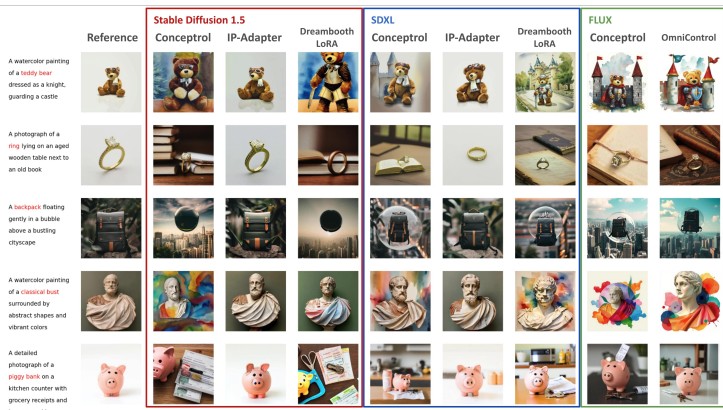

Figure 11: Auxiliary Qualitative Results. Refer to Fig. 19 and Fig. 20 for more results.

associated with reference images. Dreambooth (Ruiz et al., 2023) fine-tunes the entire diffusion model while regularizing against the hyper-class of the target to leverage the base model's semantic prior. While effective, these methods are computationally expensive. Also, the need to fine-tune a separate model for each target limits scalability, motivating the development of zero-shot adapters.

**Zero-shot Adapters** like IP-Adapter (Ye et al., 2023) and OminiControl (Tan et al., 2024) bypass fine-tuning by training additional components to incorporate image conditions directly into large-scale diffusion models. IP-Adapter (Ye et al., 2023) uses CLIP image embeddings and cross-attention layers to integrate image conditions alongside text prompts for UNet-based models. Omini-Control (Tan et al., 2024) is an adapter for DiT-based models like FLUX (Labs, 2023). It appends image condition tokens to the multi-modal attention layers and trains a LoRA adapter for efficient adaptation. These methods reduce the computational overhead, but their performance lags behind fine-tuning methods, even with large amounts of training data (Ye et al., 2023; Tan et al., 2024).

**Masked generation** use masked attention to restrict the generation within a given region (Marcos-Manchón et al., 2024; Zhou et al., 2024; Wu et al., 2023; Wang et al., 2024b; Endo, 2024). Endo (2024); Wang et al. (2024b) proposed to integrate user-defined masks, enabling precise spatial control without additional training. These approaches demonstrate that masking in the attention layer can effectively limit the generation region though using manual mask.

## 7 LIMITATION AND FUTURE WORK

Our understanding of the masking mechanism remains empirical, motivating a more theoretical study of how attention posteriors interact with adapter conditioning in the future. The method also inherits the base model's text-understanding limitations—e.g., in prompts such as "a person is standing with another person in the park," Stable Diffusion with Conceptrol may incorrectly personalize multiple subjects because CLIP treats the text as a bag of words, whereas FLUX shows fewer such failures due to stronger semantic grounding. Finally, multi-subject generation lies outside the main scope of our work; while Stable Diffusion and SDXL with Conceptrol can handle some cases, performance is less stable in challenging cases, and FLUX with OminiControl does not support multi-subject generation. Additional qualitative results are provided in Appendix C.

## 8 CONCLUSION

In this paper, we introduced **_Conceptrol_**—a simple yet effective plug-and-play method that significantly enhances zero-shot adapters for personalized image generation. Our approach is built on insights that the visual specification should be constrained by textual concept. By transferring visual specifications using the textual concept mask, Conceptrol achieves remarkable performance _without additional computation, data, or models_. Our findings underscore the importance of integrating textual concepts into personalized image generation pipelines, even with advanced architectures.

## ETHICS STATEMENT

This work advances personalized image generation by improving zero-shot adapters for text-to-image models. While these contributions support greater accessibility and efficiency in creative applications, they also carry risks of misuse, including the unauthorized replication or manipulation of personal likenesses. By lowering the barrier to producing highly personalized outputs, such methods could be exploited to generate deceptive or harmful content. We stress that our research is intended for academic and beneficial purposes, and we encourage responsible use of the technology, alongside safeguards such as watermarking, usage policies, and continued engagement with the ethical and societal implications of personalized generative models.

## REPRODUCIBILITY STATEMENT

We include all experiment details in Section 5, the Appendix B and the Appendix A for reproducibility. The details of the method are described in the Appendix C. We provide the anonymous code here and will release the code once the paper get accepted.

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

## A  ANALYSIS DETAILS

In our analysis, we investigate whether the attention maps from the base model and the adapter differ, and if they highlight the regions of interest in the generated images.

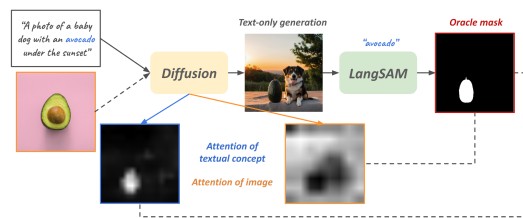

We sample 300 image-text pairs from Dream-benchPlus (Peng et al., 2024) as our analysis targets. The analysis process is shown in Fig. 12. For each generated image, we apply LangSAM (Medeiros, 2024) for open-vocabulary segmentation using the provided textual concept $c_{\text{concept}}$, referring to the resulting segmentation as the oracle mask. We then normalize the attention map obtained during image generation so that its minimum is 0 and its maximum is 1, and compute the AUC between this normalized attention map and the oracle mask. A higher AUC indicates a closer match between the attention map and the oracle mask, suggesting that the attention map accurately identifies the region of interest in the generated image. The following sections detail

Figure 12: **Analysis Process.** We generate the results fully based on the text condition. During the generation, we still compute the attention map of the reference image bet setting the conditioning scale of the reference image as 0, indicated by "$-->$" . After obtaining the text-only results, we use LangSAM (Medeiros, 2024) to retrieve the oracle mask, then compute the AUC between normalized attention map and oracle mask.

the analysis experiments. All experiments are conducted over five runs with different random seeds.

**Misaligned attention to reference images with textual concept**. As reported in Sec.3.2, the highest AUC for the image condition across all blocks averaged on analysis samples is only 0.38, whereas the highest AUC for the specified textual concept is 0.99 we also presented in Tab. 3. Qualitatively, the attention to image condition is usually globally distributed on every foreground subject, easily leading to artifacts (e.g., a dog and an avocado are rendered as two avocados).

**Visual specification can be transferred**. We further verify if manually adjusting the attention map of the image condition with the oracle mask can constrain the effect of additional image conditions. The experiment is conducted as follows: 1) Similar to previous analysis, we generate the images fully based on the given text, termed $I_{c_{\text{text}}}$ and obtain oracle mask $M_{c_{\text{concept}}}$ from it; 2) We use this oracle mask to mask the attention map of image condition, then generate with text and image together to

Table 3: **highest AUC of attention on reference image with oracle mask on Stable Diffusion 1.5.** The AUC of the textual concept is dominantly higher than the image condition.

| BLOCK NAME | textual concept | reference image |
|---|---|---|
| UP.1.1.0 | 98.89 | 37.72 |
| UP.1.2.0 | 98.49 | 34.46 |
| UP.1.0.0 | 99.15 | 24.19 |
| DOWN.2.1.0 | 95.05 | 15.66 |

get another generated image called $I_{c_{\text{fused}}}$, and obtain its oracle mask similarly $M_{c_{\text{fused}}}$; 3) We then compute AUC between $M_{c_{\text{concept}}}$ and $M_{c_{\text{fused}}}$. If this AUC is higher, it indicates that visual specification is transferred better to the intended area. In our experiment, the AUC between $M_{c_{\text{fused}}}$ and $M_{c_{\text{concept}}}$ is close to 0.9 across each base model and adapters, which strongly shows that visual specification can be transferred within regions of high attention score. As Fig. 13 shows, this can provide better results than a vanilla baseline, which is harder to maintain a balance between prompt following and

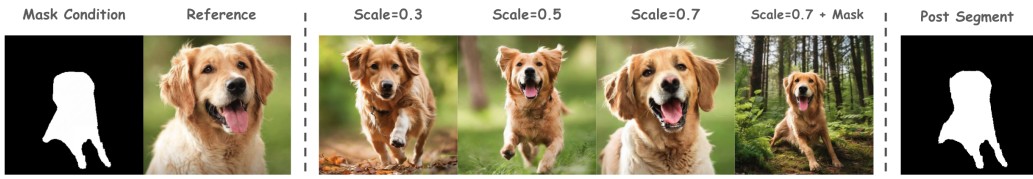

Figure 13: **Example of visual specification transferred by mask using SDXL + IP-Adapter.** We can use a manually provided mask to control the attention, thereby controlling the area of visual specification. However, this requires an additional manual mask, which requires an external model and complicated processing.

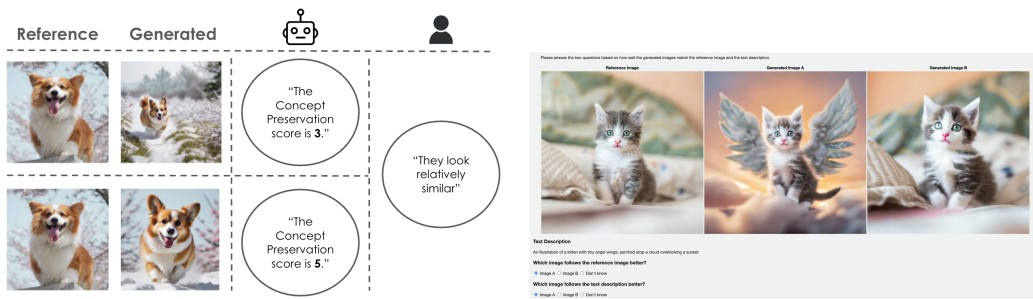

(a) **GPT-4 v.s. Human Evaluator**  (b) **Human Study Screenshot**.

Figure 14: **Human Study Details**. (a) GPT-4 may misjudge concept alignment when layout and pose differ. (b) The user is given a reference image, text prompt, and permuted image pairs generated by vanilla adapters and the version with Conceptrol. They are required to answer two questions: 1) which image preserves the concept better, and 2) which image follows the text prompt better?

concept preservation. However, a manual mask is hard to obtain in the real application, where user need to figure out the layout by themselves or require an external model. Additionally, it also faces a challenge where the personalized target is abstract, such as style.

**Concept-specific blocks for text conditions indicate the region of interest during generation**. In this analysis, we are interested in which attention block of text condition provides the highest AUC with the oracle mask. As reported in Sec . 3.3, we found that while some blocks provide less information on the region of interest (e.g., the blocks that are close to the input or output might focus more on the existence of noise), there exist blocks indicating the region of interest with a high attention score. In our implementation over Conceptrol, we use UP BLOCK 1.0.0 in Stable Diffusion, UP BLOCK 0.1.5 in SDXL, and BLOCK 18 in FLUX as the concept-specific block to extract the region of interest from the model themselves.

## B  MAIN EXPERIMENT DETAILS

**Human Study Details.** Fig. 14 (b) displays a screenshot of the survey used on Amazon Mechanical Turk. For each base model, we randomly sampled 200 pairs from the results generated by DreambenchPlus (Peng et al., 2024) and had three human annotators evaluate each comparison. The win rate reported in the main text is computed as follows: in a comparison between method A and method B, if an annotator selects method A, then method A receives a score of 1 and method B receives a score of 0; if method B is selected, the scores are reversed. When an annotator chooses "Don't know," both methods receive a score of 0.5. After processing all pairs, we calculate the win rate based on the total scores.

**Discussion over the discrepancy between GPT-4 evaluation and human evaluation.** As discussed in Sec. 5, we observe a notable discrepancy between GPT-4–based evaluation and human judgment. For models such as SD and SDXL with IP-Adapter, copy–paste artifacts often lead GPT-4 to assign higher concept-preservation scores, despite these artifacts being undesirable to human raters. Similarly, when the same identity appears with different orientations or poses, GPT-4 may assign lower scores than a human evaluator, as Fig. 14 (a) shows. This behavior is consistent with the findings reported in DreamBench, which show that GPT-4 is more reliable for assessing prompt following than for evaluating concept preservation.

**Additional Human Study on Dreambooth LoRA versus SDXL + Conceptrol.** We additionally include a comparison with DreamBooth-LoRA on SDXL, following the same settings that we compare our methods against the baseline, evaluated under the same human study setting described in the paper. As shown in Table 4, Conceptrol achieves slightly better concept preservation than Dreambooth-LoRA while maintaining the performance of prompt following, which shows consistent performance comparison over DreamBench++ and CustomConcept101.

Table 5: Results on CustomConcept101 based on Stable Diffusion.

| Method | Requires FT | Text Align (CLIP) | Img Align (CLIP) | Img Align (DINO) |
|---|---|---|---|---|
| Textual Inversion | Yes | 0.6117 | 0.7530 | 0.5128 |
| DreamBooth | Yes | 0.7514 | 0.7521 | 0.5541 |
| Custom Diffusion | Yes | 0.7583 | 0.7456 | 0.5335 |
| IP-Adapter | No | 0.6543 | **0.7907** | **0.6336** |
| IP-Adapter + Conceptrol | No | **0.7875** | 0.7684 | 0.6279 |

**More Qualitative Results**. We present additional qualitative results in Fig. 19, Fig. 20, and Fig. 11 across various customized targets—including animals, humans, objects, and styles. As shown, our method is generally as competitive as Dreambooth LoRA and often

Table 4: Human study win rates comparing DreamBooth-LoRA and Conceptrol on SDXL.

| Metric | Concept Preservation | Prompt Following |
|---|---|---|
| **DreamBooth-LoRA** | 52% | 39% |
| **Conceptrol** | 48% | 61% |

outperforms it, without any extra computational overhead and with fewer copy-paste artifacts (e.g., the corgi in the 9th row of Fig. 19). Moreover, our approach improves concept preservation for FLUX when integrated with OmniControl. For instance, in the 10th row of Fig. 19, Conceptrol successfully captures the distinctive features of a cat resembling a telephone operator. Additionally, even when FLUX is trained without human data, Conceptrol enables effective customization, as demonstrated in the 3rd–5th rows of Fig. 20 (although several failure cases are observed in the 1st–2nd rows).

**Computation Overhead.** Conceptrol does not introduce any additional inference passes. Instead, it operates entirely on the attention information already computed within the base model, reusing the concept-specific attention maps without duplicating the sampling process. This design is intentional: it enables spatially guided personalization while keeping the computational cost effectively unchanged. To quantify the overhead, we compare SDXL + IP-Adapter with and without Conceptrol using 50 inference steps (float16) on a single RTX A5000 GPU. The runtime increases only marginally (19.30 s vs. 19.53 s), confirming that the additional cost is negligible. The only extra operations performed by Conceptrol are extracting the attention-derived mask and applying it during decoding. Since the adapter already integrates text and image features through either direct addition or MM-Attention, our method does not require any auxiliary forward passes.

**Results on CustomConcept101.** To evaluate generalization beyond DreamBench++, we further test on CustomConcept101, a diverse single-subject personalization benchmark covering 101 user-defined concepts, including objects, pets, wearables, scenes, and household items. Compared with DreamBench++, which provides only a single reference image, CustomConcept101 offers 3–8 images per concept, allowing richer visual variation. For a fair comparison with prior work, we also perform our experiments using Stable Diffusion. When multiple reference images are available, we extract their image embeddings and average them as the conditioning signal. Results are shown in Table 5. IP-Adapter + Conceptrol improves textual alignment from 0.6343 to 0.7875 while maintaining competitive image alignment (CLIP/DINO), all without any fine-tuning. Compared with fine-tuned baselines such as Textual Inversion, DreamBooth, and Custom Diffusion, our method achieves comparable or even stronger alignment at zero training cost. These results support the effectiveness and generality of Conceptrol in more diverse, real-world personalization scenarios.

## C   METHOD DETAILS

We provide the overview of our method in Figure 6. Below, we introduce more details regarding the implementation of MM-Attention and the choice of warm-up ratio.

**Details of Conceptrol Warm-up**. In our experiments, we use a warm-up ratio of 0.2 for Stable Diffusion and SDXL, but 0.0 for FLUX. We observed that the area under the curve (AUC) between the textual concept mask and the ground truth mask is lower for Stable Diffusion and SDXL than for FLUX, as shown in Fig. 4 of the main text. In Fig. 4, during the first ten steps, the AUC for Stable Diffusion and SDXL remains below 0.8 for most of the time, whereas FLUX reaches an AUC greater than 0.8 at the very first step. Additionally, Fig. 17 visualizes that FLUX's textual concept mask converges much faster than those of Stable Diffusion and SDXL. Empirically, the

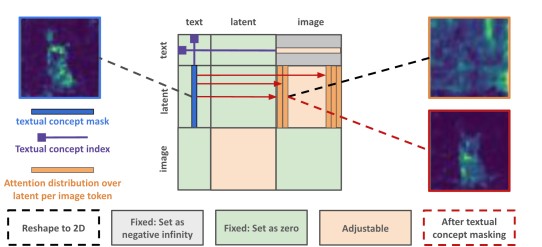

(a)          (b)          (c)          (d)          (e)

Figure 15: **An example of ablation study on Conceptrol with MM-Attention**. All images are generated with the prompt "A statue is reading in the cafe" using FLUX as the base model. From left to right, the images are (a) reference image, (b) generation without reference image, (c) with OminiControl, (d) not suppressing attention of irrelevant text tokens, and (e) with full Conceptrol.

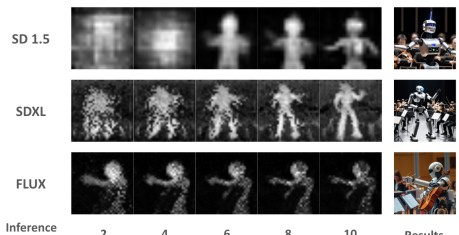

Figure 16: **Visualization of adjusted MM-Attention of Conceptrol**. We first locate the textual concept index, i.e., $[i_s, i_e]$, then extract the textual concept mask and apply it to the distribution over latent per-image tokens.

Figure 17: **Textual Concept Mask of Stable Diffusion, SDXL, and FLUX.** All images were generated using the text prompt "A robot is playing violin," with "robot" serving as the designated textual concept.

AUC of FLUX's textual concept mask typically converges by the second step (out of fifty steps), while Stable Diffusion and SDXL take about ten steps to converge. Therefore, we apply conditioning warm-up in Stable Diffusion and SDXL, which is not necessary for FLUX.

**Adaptive warm-up ratio.** The fixed warm-up ratio used in our main experiments reflects the empirical observation that attention masks gradually stabilize over the early diffusion steps. To remove this manually chosen hyperparameter, we additionally develop an adaptive warm-up strategy that monitors the convergence of the attention mask during inference. Specifically, at each timestep we compute the AUC between the current attention mask and those from previous steps, and activate Conceptrol once this AUC exceeds a predefined stability threshold. To verify the adaptive mechanism, we further conduct experiments on Stable Diffusion 1.5. As reported in Table 6, the adaptive strategy achieves performance on par with the fixed 20% warm-up configuration.

**Details of Conceptrol on MM-Attention**. The visualization of the multi-modal attention mask applied in the Conceptrol is shown in Fig. 16. For instance, given the textual prompt "a cat is chasing a butterfly" and a personalized image of a cat, we first index the text token corresponding to "cat" in the prompt, i.e., $[i_s, i_e]$ in the main text.

Table 6: Comparison between fixed warm-up (20%) and adaptive warm-up strategy on Stable Diffusion.

| Method | Concept Pres. | Prompt Follow. | Overall |
|---|---|---|---|
| IP-Adapter | 0.881 | 0.238 | 0.210 |
| IP-Adapter + Conceptrol (fixed 20%) | 0.500 | 0.795 | 0.397 |
| IP-Adapter + Conceptrol (adaptive) | 0.517 | 0.782 | 0.404 |

Then we slice the attention of latent to the tokens of textual concept and obtain textual concept mask. Each column vector in the attention of the latent to the reference image is subsequently masked with the textual concept mask. Additionally, we filter out the attention of irrelevant text tokens to the image. For instance, the text tokens of "chasing" and "butterfly" should not rely on the image token. We present an example in Fig. 15. Compare (d) with (c), the textual concept mask effectively increases concept preservation, i.e., the appearance of the statue is closer to the one presented in the reference image. Compare (e) with (d), suppressing attention of irrelevant text tokens to reference images can reduce the artifact (e.g., erasing the additional book at the bottom).

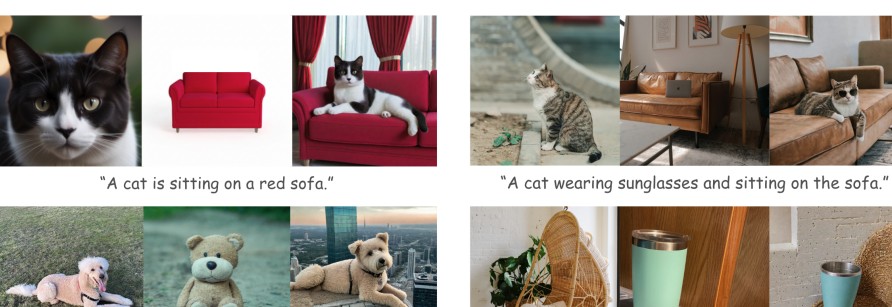

"A cat is sitting on a red sofa."    "A cat wearing sunglasses and sitting on the sofa."

"A dog lounging beside a teddy bear atop a towering skyscraper."    "A cat wearing sunglasses and sitting on the sofa."

Figure 18: **Multiple-subject customization on SDXL.** Top: successful multi-subject personalization. Bottom: typical failure cases—left, the model collapses two subjects into one; right, the chair is incorrectly customized. These cases typically require different hyperparameters for each subject for better control.

**Discussion on Multiple-subject Generation**. As demonstrated in Fig. 1 with the statue and book examples, Conceptrol offers a better ability to handle multiple distinct subjects and applying multiple personalized targets on Stable Diffusion and SDXL, as Fig. 18 shows. The qualitative result further illustrates that Conceptrol can coordinate multiple personalized concepts within a single synthesis when supported by the underlying model architecture. However, this setting is not directly applicable to FLUX when used with OminiControl, whose native control mechanism does not support multiple personalized targets. Additionally, the generation with multiple reference is relatively unstable in challenging cases requiring hyperparameter tuning for successful cases. To maintain a consistent comparison across base models, we therefore refrain from reporting quantitative results on multi-subject generation and leave a comprehensive evaluation of this scenario to future work.

**Discussion with LoRAShop.** LoRAShop (Dalva et al., 2025) is a concurrent work that extracts concept-specific masks from attention activations in a rectified-flow transformer (DiT) to enable multi-concept blending via spatially gated LoRA adapters. It requires pre-fine-tuning LoRA for each subject. In contrast, Conceptrol focuses on zero-shot personalization with a single reference image and prompt, using a textual concept mask to improve subject preservation and prompt adherence without multi-adapter blending or architecture constraints. While both leverage attention-based masks, the methods target different settings and objectives.

## D    FORMULATION DETAILS

**Text-to-Image Diffusion Models** generate images by progressively denoising a latent, $x_T$, sampled from a standard Gaussian distribution. The process is guided by a denoising function, $\epsilon_\theta$, and conditioned on a text prompt, $c_{\text{text}}$. The reverse diffusion process conditionally samples latent $x_{t-1}$, in a latent space encoded by a VAE (Kingma, 2013):

$$p(x_{t-1}|x_t, c_{\text{text}}) = \mathcal{N}(x_{t-1}; \mu_\theta(x_t, t, c_{\text{text}}), \Sigma_\theta(t)). \tag{10}$$

The mean in Equation 10, $\mu_\theta(x_t, t, c_{\text{text}})$, is computed as:

$$\mu_\theta(x_t, t, c_{\text{text}}) = \frac{1}{\sqrt{\alpha_t}}\left(x_t - \frac{\beta_t}{\sqrt{1-\bar{\alpha}_t}}\epsilon_\theta(x_t, t, c_{\text{text}})\right), \tag{11}$$

with hyperparameters $\alpha_t$ and $\beta_t$, while $\Sigma_\theta(t)$ represents a fixed or learned variance. The denoising function $\epsilon_\theta$ is represented by a UNet (Ronneberger et al., 2015; Rombach et al., 2022; Podell et al., 2023) or a ViT (Dosovitskiy et al., 2020; Labs, 2023). Starting with $x_T$, the denoising process iteratively refines the latent to reach $x_0$, which is then decoded into an image with a VAE decoder. Note that FLUX (Labs, 2023) uses flow matching (Lipman et al., 2022) with a standard Gaussian distribution, which is also considered a form of diffusion (Wang et al., 2024a).

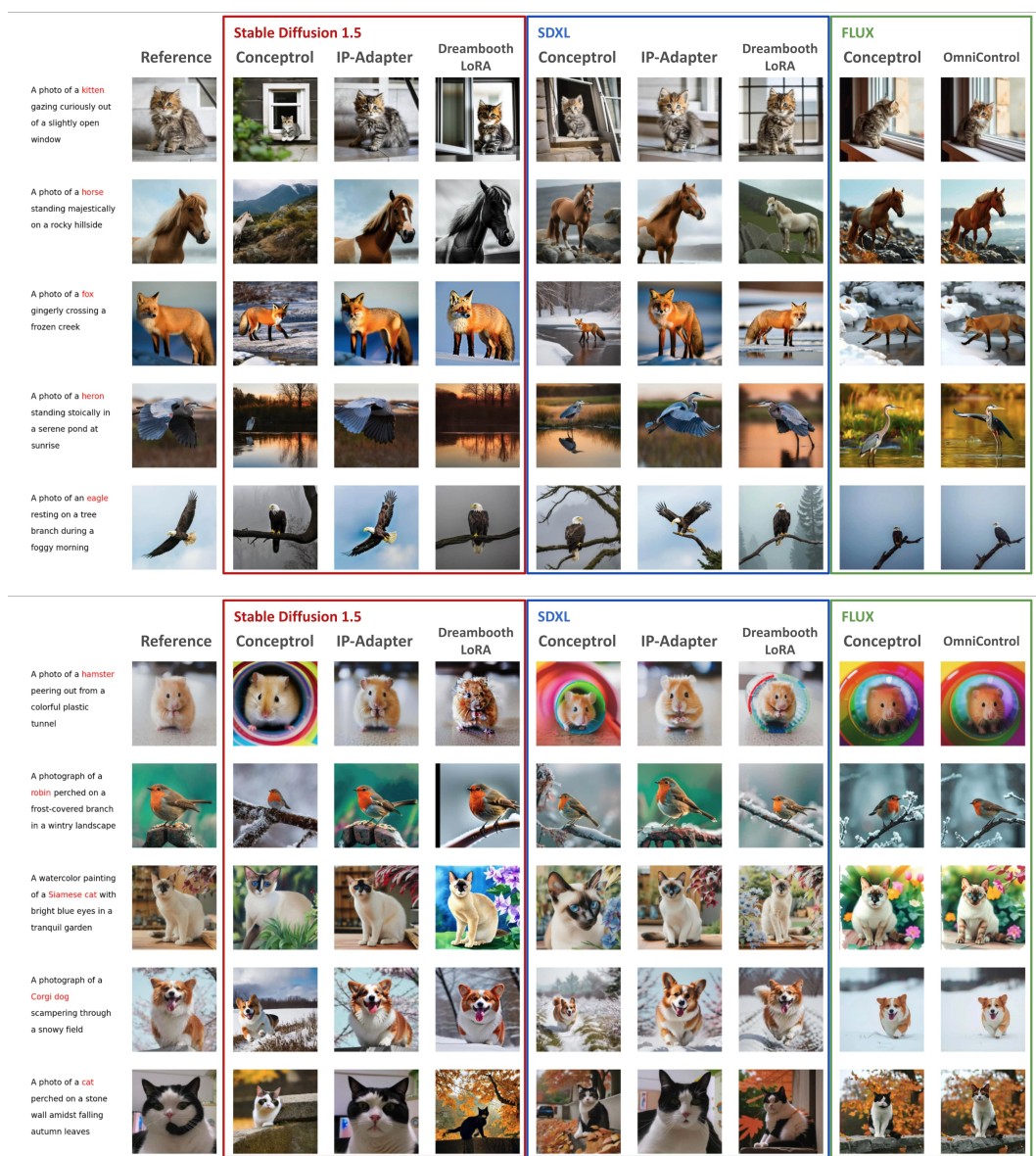

Figure 19: More Qualitative Results.

# E  THE USE OF LARGE LANGUAGE MODELS

We use a Large Language Model to aid and polish writing, e.g., grammar checking and fluency improvement.

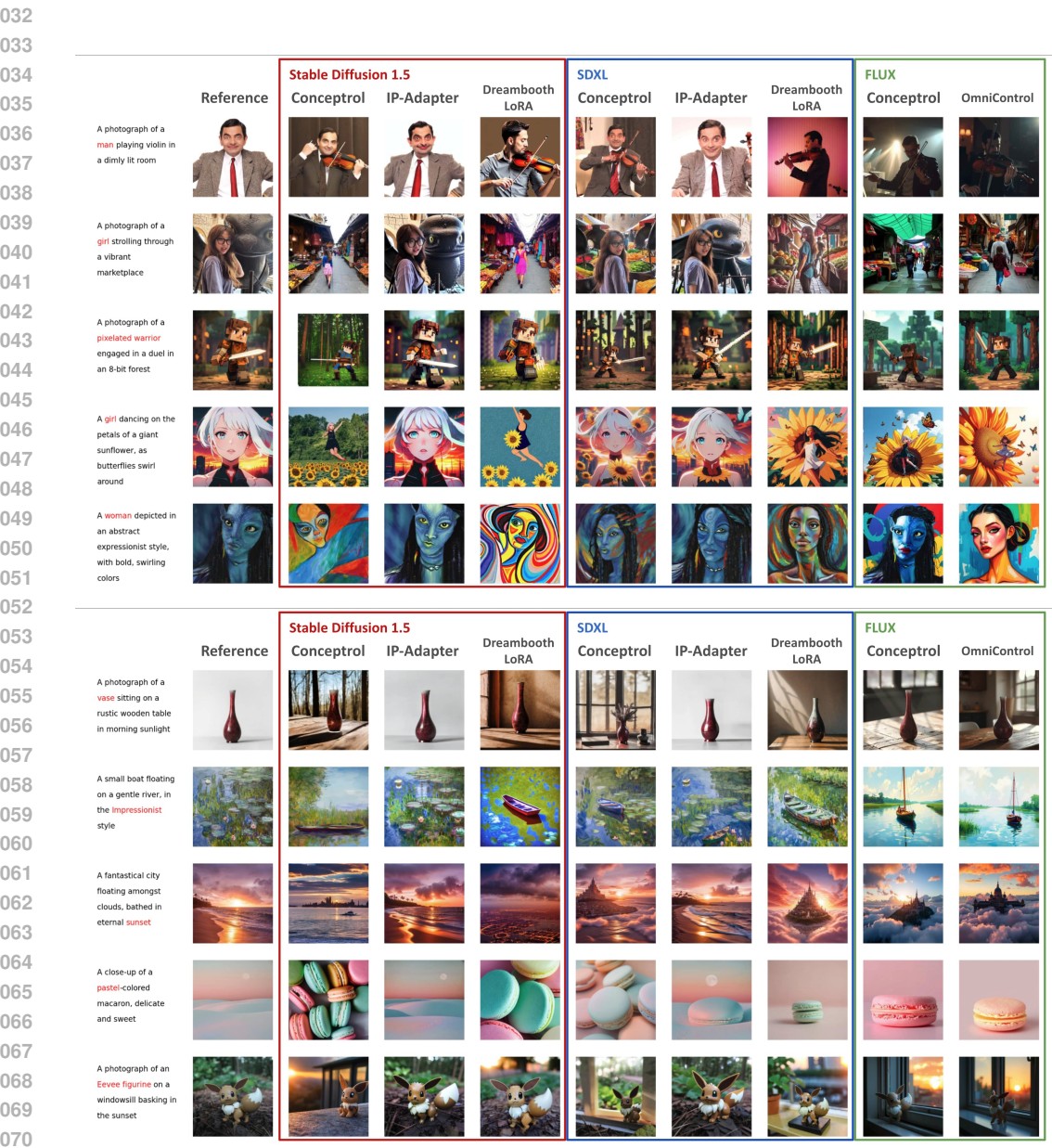

Figure 20: More Qualitative Results.

