# OpenReview forum: "Conceptrol: Concept Control of Zero-shot Personalized Image Generation"
_ICLR.cc/2026/Conference — Submitted to ICLR 2026_

### Official Review · Reviewer_LHKm · 2025-10-27

**Soundness:** 4
**Presentation:** 3
**Contribution:** 2
**Rating:** 4
**Confidence:** 5

**Summary:**

This paper focuses on the personalization task in the field of image generation. The authors analyzed the attention mechanism with image injection, and proposed a method called conceptrol to improve the performance of personalization. The experiments showed that the proposed method can achieve better performance compared to baseline methods.

**Strengths:**

- The presentation of figures is great and easy to understand.
- The evaluation is based on both UNet-based models and DiT-based models.
- The math notations in this paper are self-contained and well-defined.
- The finding that the authors claim in Sec 3.1 is insightful.

**Weaknesses:**

- I have to say that the key observations that the authors claim (Line 70-78) were found by previous research works and somehow became common sense in the image generation field. Could you bring deeper insights about these?
- Have you considered conducting a detailed evaluation of the precision of textual concept masks?
- I have to say that the current method, including extracting masks, modifying attention, and timestep warmup, lacks enough novelty. It can be considered a simple combination of existing techniques.
- The authors used an incorrect citation format in the appendices.

**Questions:**

- How many samples did you use to obtain the results in Fig. 5?
- Can you show some examples of the claim "Visual specifications can be transferred within regions of high attention score"?

(Please also see the weaknesses section for questions)

---

> ### Author Response · Authors · 2025-11-20
>
> We thank the reviewer for the suggestions on deeper discussion regarding our novelty and insights. We address these points below.
>
> > I have to say that the key observations that the authors claim (Line 70-78) were found by previous research works and somehow became common sense in the image generation field. Could you bring deeper insights about these?
> > > I have to say that the current method, including extracting masks, modifying attention, and timestep warmup, lacks enough novelty. It can be considered a simple combination of existing techniques.
>
> Our core contribution is identifying a fundamental flaw in adapter-based personalized generation: the reference image is not constrained by the textual concept. Lines 70–78 introduce this insight, which motivates our simple and effective solution.
>
> We respectfully disagree with the claim that these observations are already common knowledge. Specifically:
>
> - **Observation (1)**: Few prior works explicitly address the misalignment between adapter attention and the intended semantic region. Even recent adapter methods [1, 2] follow the paradigm tearting reference image without constraint of textual concept and still fail on challenging cases, supporting our argument in Sec. 3.1.
>
> - **Observation (2)**: While attention masking has been explored as we discussed in Sec.3.2, most prior works rely on explicit spatial masks (e.g., user-defined regions, segmentation maps, or source images). Such assumptions do not hold in zero-shot personalization, where no source image or manual annotation is available. Obtaining a suitable mask that reflects both the personalized subject and the prompt is impractical in this setting.
>
> - **Observation (3)**: To our knowledge, there is little discussion of concept-specific blocks that enable fine-grained semantic control, especially for abstract concepts such as styles.
>
> In contrast, our method extracts the attention mask on the fly from a concept-specific block, using the model’s internal attention to infer the intended generation region—without relying on post hoc inversion or full denoising. This enables effective control even for abstract or diffuse concepts (e.g., pixel-art styles) where prior mask-based approaches fall short.
>
> We would appreciate pointers to any prior works that overlap with these observations, so we can more clearly articulate the distinctions.
>
> > Have you considered conducting a detailed evaluation of the precision of textual concept masks?
>
> Thanks for this suggestion - we have! We evaluate the precision of the textual concept masks using the AUC metric (see Figure 5 in Section 3.2, Table 3 in Appendix A). It shows that textual concept mask correctly capture the intended area of corresponding concept.
>
> > The authors used an incorrect citation format in the appendices.
>
> Thank you for pointing this out. We have corrected these in the revision.
>
> > How many samples did you use to obtain the results in Fig. 5?
>
> We used 300 image–text pairs from DreamBenchPlus, evaluating each pair with five different random seeds. The results in Figure 5 are averaged over these 1,500 generations.
>
> > Can you show some examples of the claim "Visual specifications can be transferred within regions of high attention score"?
>
> Thank you for the suggestion. We added additional explanations and examples in Appendix A (Fig.13). Specifically, we tested over 300 samples by manually adjusting the attention maps with the provided mask and observing whether the personalized content appears in regions with high attention scores.
>
> **Again, we thank the reviewer for the valuable suggestions and have updated the revision (marked in red) accordingly. We hope our response addresses all concerns and welcome further questions.**
>
> [1] Diffusion Self-Distillation for Zero-Shot Customized Image Generation, CVPR 2025.
>
> [2] OminiControl: Minimal and Universal Control for Diffusion Transformer, ICCV 2025.

---

> > ### Comment · Reviewer_LHKm · 2025-11-24
> > **Response to Authors**
> >
> > Thanks authors for their response and efforts to solve my concerns.
> > However, some concerns remain about the observations in your work:
> > - Observation 1: These two works you mentioned are not the so-called *adapter* methods. They use a different method to inject visual semantics.
> > - Observation 2: I have to say that extracting and reusing spatial masks for visual concepts is indeed a normal approach in the image generation field (e.g., attention refocusing-CVPR24, FreeCustom-CVPR24). This is not saying that you cannot adopt this technique in your work, but overclaiming might not be a good way.
> > - Observation 3: "Concept-specific blocks" are also well discussed (e.g., B-LoRA-ECCV24).

---

> > > ### Author Response · Authors · 2025-11-24
> > >
> > > Thank you for the timely feedback and for outlining your concerns in detail! We address each remaining concern in the following response.
> > >
> > > ---
> > >
> > > ### Observation 1
> > >
> > > We would like to clarify that, to the best of our knowledge, adapter methods are generally defined as approaches that introduce additional parameters additional to the base model. Under this definition, both referenced works indeed rely on adapter-based mechanisms. Specifically, they use **LoRA (Low-Rank Adaptation)** [1] with additional conditioning token, commonly named as *LoRA adapters* [2–4], which adds learnable low-rank parameters to the base model to incorporate additional visual semantic tokens. This design is recognized as zero-shot adapter compared to testing-time fine-tuning methods such as DreamBooth or Textual Inversion. As demonstrated in the main paper, Conceptrol works well on OminiControl which uses LoRA adapter.
> > >
> > > ---
> > >
> > > ### Observation 2
> > >
> > > We agree that spatial masking has been explored previously, as noted in our earlier response (“attention masking has been explored as discussed in Sec. 3.2”). However, we respectfully disagree that our contribution is overstated. We do not claim that “extracting and reusing attention masks” as our contribution. Rather, our contribution lies in addressing a distinct question: *Can zero-shot adapters be guided by the textual concepts that define the visual specification?* To the best of our knowledge, prior work has not examined the interaction between adapter behavior and the base model’s attention in this context.
> > >
> > > Most prior methods depend on **explicit spatial masks** for visual concepts, such as user-defined regions, segmentation maps, or direct source-image supervision. For the cited works:
> > >
> > > - **Attention Refocusing** requires manually provided layout information (their Fig. 2) and is used for grounded text-to-image synthesis.
> > > - **FreeCustom** requires an external segmentation model (their Fig. 4) and is used without zero-shot adapters.
> > >
> > > This is exactly why our paper properly **cited that there is substantial prior literature on using attention masks for spatial control (lines 097–086 in Sec. 1; lines 514–518 in Sec. 6)**. Additionally, we also clearly state that *"it has been shown that ... by attention masking"* (lines 245-247 in Sec.3). However, such approaches are **not feasible** in the personalization setting using zero-shot adapters, where no external spatial mask or supervised spatial guidance is available. In contrast, our method focus finding the attention mask that is applicable to resolve the conflicts with textual concept and visual speficiation from the base model itself.
> > >
> > > The difference is also demonstrated by the generalization ability. For example, neither Attention Refocusing nor FreeCustom can generalize to **style-level customization**, where no explicit mask exists because the potential attention mask is globally distributed, where simple spatial masking is not applicable. Our concept mask remains effective in such cases (e.g., Fig. 7 and the “pixel-art” example in Fig. 8), .
> > >
> > > We hope this clarifies why our claims are appropriate and how our contribution differs from existing attention-masking techniques.
> > >
> > > ---
> > >
> > > ### Observation 3
> > >
> > > While B-LoRA also introduces some specific blocks, they are not claimed as concept-specific, and are also fundamentally different from our **concept-specific blocks** in three aspects:
> > >
> > > 1. **Application**
> > >    B-LoRA uses specific blocks to constrain LoRA *testing-time fine-tuning*.
> > >    Our approach uses concept-specific blocks for **training-free** personalization.
> > >
> > > 2. **Motivation**
> > >    B-LoRA is used to separate style and content in LoRA fine-tuning.
> > >    Our method identifies where an additional adapter should contribute to resolve the imbalance between textual concepts and visual specifications.
> > >
> > > 3. **Implementation Details**
> > >    In SDXL (where B-LoRA is implemented), their specific blocks correspond to `UP_BLOCK.0.0` for content customization. In our case, the concept-specific block is `UP_BLOCK.0.1.3`, which serves a different architectural function. Using the specific block of B-LoRA to guide training-free, zero-shot adapter is not effective.
> > >
> > > ---
> > >
> > > We would like to emphasize that our observations are made in the context of personalization using zero-shot adapters, whereas the works mentioned focus either on modifying only the base model (e.g., Attention Refocusing, FreeCustom) or on fine-tuning–based approaches (e.g., B-LoRA Shop). As a result, their application scope, motivation, and implementation details differ from our setting, which also makes adopting existing attention masking technique is not applicable.
> > >
> > > **We thanks the reviewer again for valuable input and we will add further detailed discussion in the revision to clarify these distinctions. We hope our response addresses the remaining concerns.**
> > >
> > >
> > > [1] LoRA, ICLR 2022.
> > >
> > > [2] LoRAverse, ICCV 2025.
> > >
> > > [3] Concept Sliders, ECCV 2024.
> > >
> > > [4] DragLoRA, ICML 2025.

---

### Official Review · Reviewer_fRKq · 2025-10-31

**Soundness:** 3
**Presentation:** 3
**Contribution:** 2
**Rating:** 4
**Confidence:** 4

**Summary:**

The paper introduces Conceptrol, a plug-and-play zero-shot framework for personalized image generation with text-to-image diffusion models. Unlike fine-tuning-based methods such as DreamBooth or LoRA, Conceptrol enhances zero-shot adapters (e.g., IP-Adapter, OmniControl) by constraining visual attention through a textual concept mask. This aims to better balance content preservation and adherence to text prompts. The approach requires no additional computation or training, yet shows up to 89% improvement over IP-Adapter on personalization benchmarks and even surpasses fine-tuned methods in some cases.

**Strengths:**

- Clear motivation and intuition: the paper convincingly explains why the proposed concept-mask mechanism should work. The intuition connecting attention alignment and subject-driven generation is well presented and easy to follow.
- Simplicity and generality: the approach is lightweight, requires no training, and can be easily integrated into existing diffusion pipelines.
- Significant improvement based on quantitative analysis: the method outperforms both adapters and even training-dependent approaches like DreamBooth-LoRA.

**Weaknesses:**

- Runtime and efficiency claims

Although Conceptrol is presented as zero-shot, it may internally require multiple inference passes (e.g., text-only and text+image). The paper claims “no computational overhead”, but this is not empirically substantiated.

- Inconsistent evaluation results

Quantitative concept preservation metrics drop substantially (up to 50–80%) relative to IP-Adapter on SD and SDXL, yet the user study reports only a small degradation. This inconsistency is not discussed and undermines confidence in the evaluation.

- Comparison completeness

User studies are conducted only against adapter-based baselines, while fine-tuned methods such as DreamBooth and LoRA are omitted. This limits the interpretability of the results, especially given the observed divergence between quantitative and perceptual evaluations.

- Qualitative analysis placement and fairness

The qualitative analysis is largely moved to the appendix, leaving the main paper focused on quantitative discussion. Key qualitative comparisons should appear in the main text.
Moreover, the IP-Adapter results shown are unexpectedly poor compared to the original paper, where it produced coherent images rather than simply copying the reference. This discrepancy raises concerns about possible cherry-picking or unfair settings.

- Writing

The introduction and methodology are overly long and repetitive. The discussion of “block 18” in Flux also overlooks relevant prior work (e.g., LoRAShop).

**Questions:**

- Inference efficiency

Please report the exact inference time per image and compare it with IP-Adapter and LoRA-based methods to clarify whether the “no computational overhead” claim holds in practice.

- Evaluation discrepancy

Why do the quantitative and user study evaluations diverge so strongly?

Are the quantitative metrics (e.g., concept preservation) truly aligned with human perception?

- Comparison baselines

Could you include fine-tuned baselines (DreamBooth, LoRA) in user study results?

Were all visualizations and metrics computed under identical sampling parameters (seed, steps, CFG scale)?

- Teaser figure clarity

The dashed lines and columns in the teaser figure are unclear.

What exactly is used as input in each column — only the book, only the monument, or both?

Please annotate or relabel the figure for clarity.

- Qualitative analysis

Please move key qualitative examples to the main text.

For each image prompt, it would be informative to show multiple generations with different textual prompts to demonstrate stability.

---

> ### Author Response · Authors · 2025-11-20
>
> We thank the reviewer for comprehensive review and valuable suggestions. We appreciate the recognition that our method is well-motivated and simple with significant experimental improvement and generalization ability. Below we address your concern one-by-one.
>
> > Runtime and efficiency claims: Although Conceptrol is presented as zero-shot, it may internally require multiple inference passes (e.g., text-only and text+image). The paper claims “no computational overhead”, but this is not empirically substantiated.
>
> Thank you for the question. We would like to clarify that Conceptrol does not introduce any additional inference passes. This is a key motivation for leveraging the model’s internal attention masks: they allow us to guide personalization without duplicating the sampling process.
>
> For reference, the computation cost of SDXL + IP-Adapter and its enhanced version with Conceptrol (50 inference steps, float16, single RTX A5000) is summarized below:
>
> | Method | GPU | Inference Steps | Time Cost (s) |
> | --- | --- | --- | --- |
> | SDXL + IP-Adapter | RTX A5000 | 50 | 19.30 |
> | SDXL + Conceptrol | RTX A5000 | 50 | 19.53 |
>
> The overhead introduced by Conceptrol is negligible, as the only extra operations involve extracting the concept-specific attention mask and applying it during decoding. We add the additional dicussion on the computation overhead in the revision.
>
> Finally, note that the adapter already processes text and image features jointly—either through direct addition or MM-Attention. Conceptrol does not require any additional forward passes; it simply reuses the attention information already computed by the base model. We hope this clarifies the concern regarding computational cost.
>
> > Inconsistent evaluation results ... evaluation.
>
> We discuss this observation in Sec.. 5.2 [improvement over human preference] (line 391 - 405). Specifically, IP-Adapter on SD and SDXL suffers from a copy-paste issue; GPT-4 consistently gives the highest scores for these copy-paste samples. Additionally, regarding the same identity with a different facing direction, it might not be assigned as high a score as the one that is fully identical. This also aligns with the DreamBench paper's observations that GPT-4 is more reliable for evaluating prompt following rather than concept preservation. We will add more details in Appendix A in the revision.
>
> > Comparison completeness: User studies are conducted ... between quantitative and perceptual evaluations.
>
> We add a comparison with DreamBooth-LoRA on SDXL, evaluated under the same human study setting described in the paper. The results are summarized below:
>
> | Mehod | Concept Preservation | Prompt Following |
> | --- | --- | --- |
> | DreamBooth LoRA | 52% | 39% |
> | Conceptrol | 48% | 61% |
>
> This comparison has been added in the revised version. Conceptrol achieves slightly better concept preservation than Dreambooth-LoRA while maintaining the performance of prompt following, which shows consistent performance comparison over DreamBench++ and CustomConcept101.
>
> > Qualitative analysis ... unfair settings.
>
> Thank you for this point. We were severely constrained by page limits. In the revision, we have moved several examples from the supplementary into the main paper for clarity, as suggested.
>
> We appreciate the reviewer’s concerns. However, **we respectfully disagree with this characterization.** As detailed in the paper, all experiments strictly follow the official DreamBench protocol. The IP-Adapter results are directly reproducible and can also be verified using the publicly available samples on the DreamBench project page.
>
> We also point the reviewer to qualitative results, including Fig.11, Fig.19, and Fig.20, which include several failure cases of Conceptrol (the horse in Fig. 19, the girl in Fig. 20). These qualitative results are intentionally uncurated for transparency.
>
> Lastly, we provide full anonymous code to support direct testing and independent verification. We hope this addresses concerns of fairness and completeness.
>
> > Writing: The introduction ... relevant prior work (e.g., LoRAShop).
>
> Thank you for the helpful suggestions. We will streamline the introduction and methodology. Regarding LoRAShop, we appreciate the reviewer pointing out this relevant concurrent work. It became available only after our submission, and we have now added a discussion of it in Appendix C.
>
> > Teaser figure clarity ... relabel the figure for clarity.
>
> Thanks for the suggestions. In the first segment (left of the first dashed line), only the book is personalized. In the second segment (middle), only the statue is personalized. In the third segment, both the statue and the book are personalized. We have updated the teaser in the revised version.
>
> **Once again, we thank the reviewer for the valuable suggestions and have updated the revision accordingly. We welcome further questions.**

---

> > ### Comment · Reviewer_fRKq · 2025-11-25
> > **Response to Authors**
> >
> > Thank you for providing the runtime measurements of your method. It is now clear that the method indeed does not require any additional forward passes, and that the overhead introduced by Conceptrol is negligible. The intuition explaining the discrepancy between the user study and the quantitative analysis is clear and well articulated in the appendix. I also appreciate the additional user study, and the updated teaser has become significantly more informative.
> >
> > However, I still have one remaining question: am I correct in understanding that your method does not perform well for multi-concept generation on Flux, and that this is why such results are not included in the teaser?

---

> > > ### Author Response · Authors · 2025-11-27
> > >
> > > We sincerely appreciate your continued engagement and your helpful suggestions, especially regarding additional human studies and inference-time analysis, which have strengthened our work. FLUX was not included for multi-subject generation due to limitations of OminiControl, which is unlike IP-Adapter, does not natively support multiple personalization targets; thus, we report multi-subject results only with IP-Adapter and have clarified this in Section 7 of the revision.
> > >
> > > As the discussion period concludes in the coming days, we would be grateful to know whether the updates address your concerns. We remain happy to provide any further clarification.

---

### Official Review · Reviewer_qD1L · 2025-11-01

**Soundness:** 3
**Presentation:** 3
**Contribution:** 3
**Rating:** 6
**Confidence:** 3

**Summary:**

The paper proposes Conceptrol, a training-free, inference-time control method that improves zero-shot personalized image generation with diffusion adapters (e.g., IP-Adapter for U-Net, OminiControl for DiT/FLUX). The key idea is to derive a textual concept mask from specific attention blocks in the base model during generation and use it to modulate the adapter’s image-conditioned attention, constraining the visual specification to the intended region of the textual concept. The method requires no fine-tuning, no auxiliary segmentation models, and negligible computational overhead.

**Strengths:**

1 Clear insight and elegant solution: converting textual attention into a concept mask and using it to spatially modulate image conditioning is simple, principled, and practical.

2 Training-free and model-agnostic: works with both U-Net and DiT (FLUX) stacks, requires no extra models, and adds negligible overhead.

3 Strong empirical improvements: large gains on DreamBench++ for both concept preservation × prompt following; competitive or better than fine-tuning methods while remaining zero-shot.

**Weaknesses:**

1 Early-step reliability: The warm-up mitigates unreliable early attention, but the method’s sensitivity to the warm-up ratio and inference schedule is dataset/model dependent; more systematic guidance would help.

2 Segmentation-free but not fully layout-aware: While the approach avoids external masks, it still relies on the base model’s attention being spatially meaningful. Complex layouts or heavily compositional prompts may yield suboptimal masks.

**Questions:**

1 Robustness: How sensitive is the method to mis-specified concepts (e.g., “cat” vs. “kitten”) or ambiguous/nested concepts (“red leather book”)? Can you back off to multiple candidate spans?

2 Block selection: Can you propose a general diagnostic to auto-select concept-specific blocks at inference (e.g., by measuring attention-mask sparsity/contrast vs. running time)? How does performance change if the chosen block is perturbed?

---

> ### Author Response · Authors · 2025-11-20
>
> We sincerely thank the reviewer for the recognition of our contribution and the careful review! We appreciate that the reviewer considers our solution to be insightful and elegant while providing strong empirical improvement. Below, we address your concern separately.
>
> > Early-step reliability:  ... systematic guidance would help.
>
> Thank you for the helpful suggestion. We provide details of the warm-up ratio in Appendix C and illustrate the evolution of the attention mask across models and timesteps in Figure 15. To establish a more systematic guideline, we evaluated over 1500 samples per model and computed the AUC between the textual-concept attention mask and the target region in the generated image (Figure 5), which allows us to estimate the average convergence timestep.
>
> Following your suggestion, we implemented an adaptive warm-up mechanism. At each timestep, we compare the current attention mask with those from earlier steps; once the AUC exceeds a predefined threshold, we begin applying Conceptrol. This procedure achieves performance comparable to the fixed 20% warm-up:
>
> | Method                   | Concept Preservation     | Prompt Following | Overall |
> |--------------------------|----------------------------|---------------------------|-------------------------|
> | IP-Adapter       | 0.881                    | 0.238                  | 0.210                   |
> |        IP-Adapter + Conceptrol (fixed at 20% inference)       | 0.500                    | 0.795                  | 0.397                   |
> | IP-Adapter + Conceptrol (adaptive)       | 0.517                    | 0.782                  | 0.404                   |
>
> We include this adaptive version in Appendix C of the revised manuscript.
>
>
> > Segmentation-free but not fully layout-aware: While the approach avoids external masks, it still relies on the base model’s attention being spatially meaningful. Complex layouts or heavily compositional prompts may yield suboptimal masks.
>
> Thanks for the questions. We add related discussion in Section 7 since the method relies on the base models' understanding ability. Following your suggestions, we further provide qualitative results regarding multiple subject-driven generation for compositional prompts in Fig.18.
>
> > Robustness: How sensitive is the method to mis-specified concepts (e.g., “cat” vs. “kitten”) or ambiguous/nested concepts (“red leather book”)? Can you back off to multiple candidate spans?
>
> Thank you for the insightful question. In practice, the method is only mildly affected by minor mis-specifications (e.g., “cat” vs. “kitten”), but can be more sensitive to ambiguous or overly specific concept phrases. For example, if the customized target is a yellow book but the textual concept is written as “red leather book,” the model may become uncertain about whether attributes such as color should also be personalized. Empirically, we observe that FLUX is more robust to such ambiguity—likely due to its stronger text encoder—which is consistent with its higher prompt-following performance on DreamBench++. We added a related discussion in the limitation section in the revision.
>
> > Block selection: Can you propose a general diagnostic to auto-select concept-specific blocks at inference (e.g., by measuring attention-mask sparsity/contrast vs. running time)? How does performance change if the chosen block is perturbed?
>
> Thank you for the valuable suggestion. Different from the warm-up ratio, the concept-specific block is stable to have the highest AUC across different samples. While it's possible to measure attention-mask sparsity at inference time, it might be more challenging to handle the abstract textual concept, such as style. Therefore, we choose to fix the concept-specific blocks.
>
> For the perturbing chosen block, here we additionally provide the ablation study on adding standard Gaussian noise into the feature generated by the chosen block using CustomConcept101. The result is as follows:
>
> | Method                   | Requires Fine-tuning?     | Textual-alignment (CLIP) | Image-alignment (CLIP) | Image-alignment (DINO) |
> |--------------------------|----------------------------|---------------------------|-------------------------|--------------------------|
> | IP-Adapter               | No     | 0.6343 | 0.7907 | 0.6636  |
> | IP-Adapter + Conceptrol  | No         | 0.7875                    | 0.7684    | 0.6279  |
> | IP-Adapter + Perturbed Conceptrol  | No         | 0.7352    | 0.6599   | 0.4127   |
>
> This indicates the importance of the concept block. Regarding the ablation on different choices of concept block, we provide ablation study details in Tables 2 and 3.
>
> **Again, we thank the reviewer for the valuable suggestions and have updated the revision (marked in red) accordingly. We hope our response addresses all concerns and welcome further questions.**

---

### Official Review · Reviewer_SzLs · 2025-11-01

**Soundness:** 4
**Presentation:** 3
**Contribution:** 3
**Rating:** 4
**Confidence:** 4

**Summary:**

The paper introduces Conceptrol, a training-free method to improve zero-shot personalized image generation. It uses a textual concept mask, extracted from specific attention blocks of diffusion models, to guide reference-image attention and better align identity preservation with prompt adherence. The method applies to IP-Adapter and OminiControl without retraining and shows consistent gains on DreamBench++.

**Strengths:**

- The motivation is clear and relevant to zero-shot personalization.

- The approach is simple, efficient, and architecture-agnostic.

- Empirical results demonstrate strong improvements, sometimes surpassing fine-tuned baselines, with convincing qualitative examples and human studies. The paper is well-written and experimentally thorough.

**Weaknesses:**

- The conceptual novelty is limited—attention masking and semantic control have been explored before.

- The method mainly refines existing adapters rather than introducing new principles.

-  Evaluation focuses on DreamBench++ without diverse or real-world tests, and generalization across models is not well analyzed.

- The theoretical explanation of why the mask works remains shallow.

**Questions:**

1. Can the authors clearly differentiate Conceptrol from prior works on attention masking and diffusion control? What is the key conceptual advance beyond identifying “concept-specific blocks”?
2. How does the method behave with ambiguous or abstract concepts (“freedom,” “dreamlike style”) where the target region is diffuse or overlaps with the background? Are there cases where the concept mask misaligns or suppresses important context?
3. Can Conceptrol handle prompts with multiple distinct subjects (e.g., “a cat sitting on a red sofa”) when more than one concept is associated with reference images?

---

> ### Author Response · Authors · 2025-11-20
>
> We thank the reviewer for the careful and constructive evaluation. We appreciate the recognition that our method is simple, efficient, and effective, as well as the strong scores given for soundness, presentation, and overall contribution. Below, we address your concerns separately.
>
> > W1, W2, Q1: Novelty and distinction with existing methods
>
> Thank you for this question. While attention masking and semantic control have been explored, our method differs substantially in both motivation and mechanism:
>
> **Motivation — A new principle for zero-shot adapters**: Existing subject-driven generation treats the reference image and text symmetrically. Our key insight is that reference-image conditioning must be guided by the textual concept rather than applied uniformly. This introduces a new, architecture-agnostic design principle (applicable to both UNet and DiT) that yields large gains without any training.
>
> **Masking mechanism — No reliance on source masks**: Existing masking or semantic-control approaches assume image-editing settings where the source image provides spatial supervision, which do not exist in personalized generation. We instead derive masks directly from the model’s attention maps, eliminating inversion or full denoising steps and enabling personalization for abstract concepts (e.g., styles) where no spatial mask is available.
>
> **Semantic control — Fully training-free**: Because no source masks exist, prior semantic-control approaches require extra training. Our method remains entirely training-free by using the base model’s attention to regulate the adapter during subject-driven generation, which is fundamentally different from prior mechanisms.
>
> In summary, although Conceptrol plugs into existing adapters, its guiding principle, masking strategy, and training-free control mechanism are new in the context of subject-driven generation. We will clarify this further in the revision.
>
> > Evaluation focuses on DreamBench++ without diverse or real-world tests, and generalization across models is not well analyzed.
>
> We add results on CustomConcept101:
>
> | Method                   | Training Required    | CLIP-Text | CLIP-Image | DINO-Image |
> |---|---|---|---|---|
> | Textual Inversion        | Y      | 0.62                   | 0.75                  | 0.51                   |
> | DreamBooth               | Y        | 0.75                   | 0.75                  | 0.55                   |
> | Custom Diffusion         | Y                        | 0.76                    | 0.75                  | 0.53                   |
> | IP-Adapter               | N     | 0.63     | 0.79                  | 0.66                   |
> | Conceptrol  | N         | 0.79    | 0.77     | 0.63    |
>
> Regarding generalization across models, our method transfers well to three base architectures: Stable Diffusion, SDXL, and FLUX (see Table 1). In general, Conceptrol is applicable as long as the underlying text-to-image model supports attention-mask–based control. We welcome further suggestions to showcase generalization.
>
> > The theoretical explanation of why the mask works remains shallow.
>
> Thank you for raising this point. Our current explanation focuses on empirical evidence (Sec. 3.2). Intuitively, Conceptrol "reweights" the adapter’s conditioning using the model’s internal posterior over concept locations, instead of assuming a uniform spatial prior as in existing adapters.
>
> We would appreciate suggestions on specific types of theoretical analysis to strengthen our paper, and we are happy to incorporate them.
>
> > Q2: abstract concept
>
> Thanks for this great question!  We provide qualitative results on abstract concepts such as “low-poly geometric” (Fig. 1), “pixel-art style” (Fig. 7), and “impressionist” (Fig. 18). Our method remains effective for these non-literal concepts. For better presentation, we add additional qualitative results in Fig.7.
>
> A limitation arises when the base model cannot capture important contextual distinctions. For example, in the prompt “a person is standing with another person in the park,” Stable Diffusion combined with Conceptrol may personalize both individuals, because its text  encoder (CLIP) treats the prompt as a “bag of words.” In contrast, FLUX+Conceptrol rarely has this issue because its text encoder has stronger semantic grounding. We add this discussion to Sec.7.
>
> > Q3: multiple subject-driven generation?
>
> Yes, it can.  For example, in Fig. 1, Conceptrol can handle multiple distinct subjects (the statue and the book) and can apply multiple personalized targets on both Stable Diffusion and SDXL. We add the related discussion in Sec.. 7. We also add additional qualitative results, including successful and failure cases (e.g., “a cat sitting on a red sofa”) in Fig.18.
>
> **Once again, we thank the reviewer for the valuable suggestions and have updated the revision (marked in red) accordingly. We hope our response addresses all concerns and welcome further questions.**

---

### Author Response · Authors · 2025-11-20
**Common Response**

Dear Reviewers and AC,

We sincerely appreciate your time and thoughtful feedback on our manuscript. We thank all reviewers for their constructive suggestions. In response, we have carefully revised and strengthened the paper. The manuscript now includes:

- Human preference comparison between DreamBooth LoRA and Conceptrol on SDXL (Appendix B)

- Additional evaluation on CustomConcept101 (Appendix B)

- Experiments on adaptive warm-up ratio (Appendix C)

- Measured computational overhead of Conceptrol (Appendix B)

- Clearer distinction from existing attention-masking techniques (Sec. 4)

- Limitation and Future Work (Sec. 7)

- Further clarification of evaluation settings (Appendix B)

- Discussion of concurrent related work (Appendix C)

For ease of review, important revisions are temporarily highlighted in red, and minor issues are simply fixed. We hope these updates clarify our contributions and improve the overall presentation. We also invite reviewers to evaluate the provided anonymous code in the Reproducibility Section for testing the generalization ability of our method.

Thank you again for your time and consideration.

Sincerely,
The Authors

---

### Author Response · Authors · 2025-11-27

Dear Reviewers,

With the discussion period nearing its end, we would like to thank all reviewers for their thorough evaluations. We kindly invite the reviewers to let us know if we have addressed their concerns. We will provide further clarifications if needed.

Best Regards,

The Authors

---

### Author Response · Authors · 2025-12-03

Dear Area Chair,

Thank you for taking on our submission after the incident. Below is a concise summary of the paper, the reviewers’ key recognitions, and the essential updates made during the discussion period before the OpenReview incidient.

---

### Summary of Our Work

We propose **Conceptrol**, a **training-free, plug-and-play** framework for zero-shot personalized image generation using diffusion adapters (IP-Adapter, OminiControl).

Problem: zero-shot adapters treat image and text symmetrically, causing over-personalization or copy-paste artifacts.

Solution:
- Extract a **textual concept mask** from specific attention blocks during inference.
- Use it to **spatially modulate** the adapter’s image-conditioned attention.
- Requires **no training**, **no segmentation**, **no extra forward passes**, and adds **negligible overhead**.

Conceptrol works across **Stable Diffusion, SDXL, and FLUX**, showing strong improvements on **DreamBench++** and **CustomConcept101**, often matching or surpassing DreamBooth/LoRA while remaining zero-shot.

---

### Reviewers’ Positive Feedback

Across reviews, the strengths highlighted include:

- **Clear Insight and Motivation**: A simple and principled idea—using a textual concept mask derived from internal attention to guide zero-shot adapters—with a clear link to subject-driven generation.
- **Training-Free and Model-Agnostic Design**: An **architecture-agnostic** approach that works for both U-Net and DiT stacks, requires no extra models, and adds **negligible inference overhead**.
- **Strong Empirical Performance**: **Large gains on DreamBench++** in concept preservation × prompt following, often:
  - Outperforming baseline adapters without additional cost,
  - **Matching or surpassing fine-tuned methods** (e.g., DreamBooth LoRA),
  - Generalizing across **Stable Diffusion, SDXL, and FLUX**.
- **Good Presentation and Analysis**: A clear problem statement and methodology with **thorough ablations** and **well-designed comparisons**.

These points indicate that Conceptrol is a practical and effective advance for zero-shot personalization.

---

### Key Clarifications and Additions During Discussion

### 1. Novelty and relation to prior work
- Clarified distinction from attention-masking/semantic-control methods:
  - No spatial supervision; masks derived solely from internal attention.
  - Works for abstract concepts with no obvious spatial region.
  - Regulates adapters *at inference* rather than training new ones.
- Added comparisons with concurrent works (LoRAShop, FreeCustom, B-LoRA).

### 2. Additional benchmarks & generalization
- Added **CustomConcept101** results → strong CLIP/DINO scores.
- Added **human study vs DreamBooth-LoRA** → comparable or better concept preservation, stronger prompt following.
- Clarified FLUX multi-subject limitation is from **OminiControl**, not Conceptrol.

### 3. Warm-up strategy & mask reliability
- Original submission already includes **mask evolution + AUC analysis** and  **adaptive warm-up** triggered by mask reliability.

### 4. Runtime and overhead
- Reported inference time: **19.30 s → 19.53 s** (SDXL + IP-Adapter, 50 steps, A5000) and confirms **negligible practical overhead**.

### 5. Quantitative vs human-study mismatch
- Explained that identity metrics may reward copy-paste artifacts; humans prefer Conceptrol when pose/view changes.

---

We hope this summary clearly outlines the contribution of Conceptrol, the reviewers’ recognition, and the improvements made during the discussion phase. **Thank you again for your time and for taking on this assignment under challenging conditions.**

Best Regards,
The Authors

---

### Meta-Review · Area_Chair_VqXw · 2026-01-07

**Summary:**

The reviewers agree that the paper is well written, carefully executed, and shows strong empirical improvements for zero-shot personalization. Several practical concerns raised in the reviews—such as runtime overhead, evaluation inconsistencies, and early-step attention reliability—were convincingly addressed in the rebuttal.

However, the core concern around conceptual novelty remains unresolved. Multiple reviewers continue to view the method as a relatively incremental combination of existing attention masking and control techniques, and the rebuttal, while detailed, did not fundamentally change this perception. In addition, the theoretical understanding of why the proposed mechanism works remains shallow and largely empirical. Some limitations in handling complex compositional prompts and multi-concept scenarios, especially on FLUX, further narrow the scope of the claimed generality.

Overall, while the work is solid and practically useful, the remaining concerns about novelty and depth weigh heavily in the final recommendation, leading to a reject decision.

**Reviewer Concerns:**

Reviewer Concerns

Addressed by the rebuttal

Runtime and efficiency: Clearly addressed with concrete timing results and confirmation of single-pass inference.

Evaluation consistency: The discrepancy between automatic metrics and human preference is well explained and supported by added human studies, including a comparison with DreamBooth-LoRA.

Early-step attention reliability: Addressed with mask evolution analysis and an adaptive warm-up strategy.

Evaluation coverage: Partially addressed by adding CustomConcept101 and results across SD, SDXL, and FLUX.

Still outstanding

Core novelty: For some reviewers, the method is still seen as closely related to prior attention masking and control techniques; the rebuttal clarifies positioning but does not fully resolve this concern.

Theoretical depth: The explanation of why the method works remains mostly empirical.

Complex compositional scenarios: Handling of highly compositional or ambiguous prompts is still limited.

Multi-concept generation on FLUX: Remains unsupported due to limitations of OmniControl.

**Reviewer Scores:**

Reviewer SzLs
Original: 4
After discussion: 4
Rationale: While the rebuttal is detailed and adds supporting experiments, the reviewer’s main concerns on novelty and theoretical depth remain. No explicit signal that the score would increase.

Reviewer qD1L
Original: 6
After discussion: 6
Rationale: The reviewer was already positive. The rebuttal addressed technical questions (warm-up, robustness) well, but there is no indication they would further raise the score.

Reviewer fRKq
Original: 4
After discussion: 6
Rationale: The reviewer explicitly acknowledged that key concerns were resolved (runtime, evaluation mismatch, added user study) and their follow-up comments clearly indicate increased confidence.

Reviewer LHKm
Original: 4
After discussion: 4
Rationale: Despite extensive rebuttal and clarification, the reviewer maintained strong skepticism about novelty and prior-art overlap, with no indication of softened concerns.

Score Summary:
Final scores: 4, 6, 6, 4
Average score: 5.0

---

### Decision · Program_Chairs · 2026-01-26

Reject